# Higher Order Transformers With Kronecker-Structured Attention

**Soroush Omranpour**                                    *soroush.omranpour@mila.quebec*
*Mila*
*McGill University*

**Reihaneh Rabbany**                                    *reihaneh.rabbany@mila.quebec*
*Mila, CIFAR AI Chair*
*McGill University*

**Guillaume Rabusseau**                                    *rabussgu@mila.quebec*
*Mila - DIRO, CIFAR AI Chair*
*University of Montreal*

**Reviewed on OpenReview:** *https://openreview.net/forum?id=QNOaXcKFkT*

## Abstract

Modern datasets are increasingly high-dimensional and multiway, often represented as tensor-valued data with multi-indexed variables. While Transformers excel in sequence modeling and high-dimensional tasks, their direct application to multiway data is computationally prohibitive due to the quadratic cost of dot-product attention and the need to flatten inputs, which disrupts tensor structure and cross-dimensional dependencies. We propose the Higher-Order Transformer (HOT), a novel factorized attention framework that represents multiway attention as sums of Kronecker products or sums of mode-wise attention matrices. HOT efficiently captures dense and sparse relationships across dimensions while preserving tensor structure. Theoretically, HOT retains the expressiveness of full high-order attention and allows complexity control via factorization rank. Experiments on 2D and 3D datasets show that HOT achieves competitive performance in multivariate time series forecasting and image classification, with significantly reduced computational and memory costs. Visualizations of mode-wise attention matrices further reveal interpretable high-order dependencies learned by HOT, demonstrating its versatility for complex multiway data across diverse domains. The implementation of our proposed method is publicly available at `https://github.com/s-omranpour/HOT`.

## 1 Introduction

Multiway data, represented as multidimensional arrays or tensors, are pervasive across scientific, engineering, and real-world applications. In climate modeling, multidimensional time series capture spatiotemporal variations for weather and climate prediction (Nguyen et al., 2023). Medical imaging uses 3D modalities like MRI and CT scans to reveal anatomical details beyond traditional 2D images (Yang et al., 2023). Recommendation systems model user-item interactions alongside contextual factors (e.g., time, location) naturally as tensors (Frolov & Oseledets, 2016). Such examples underscore the ubiquity of multiway data and the need for models that capture complex interdependencies efficiently.

Tensors generalize matrices to higher dimensions, enabling structured representation and analysis of multidimensional relationships. They are integral in domains ranging from quantum physics (Montangero, 2018) and algebraic geometry (Landsberg, 2012) to data science and machine learning, where they encode images, videos, medical scans, and more. Tensors facilitate higher-order correlations, multiway clustering, and

dimensionality reduction via decompositions (Lu et al., 2011). Techniques such as tensor component analysis (Lu et al., 2003), dictionary learning (Bahri et al., 2019), and tensor regression (Guo et al., 2012) extend matrix-based methods, mitigating the curse of dimensionality and enabling efficient high-dimensional modeling. Among the most powerful tools for multiway data are tensor factorizations like Canonical Polyadic (CP) and Tucker decompositions, which generalize PCA and SVD to higher-order settings. These methods extract latent patterns and have broad applications in signal processing (e.g., noise reduction, EEG and fMRI analysis), computer vision (e.g., compression, recognition), and machine learning (e.g., feature learning in recommendation systems, NLP, and social network analysis) (Lu et al., 2011). In fields such as chemometrics and neuroscience, tensor factorization uncovers hidden structures and enables scalable analysis of complex data.

Despite these advances, applying modern deep learning architectures to multiway data remains challenging. Transformers (Vaswani et al., 2017) have revolutionized sequence modeling in vision (Dosovitskiy et al., 2020), speech (Dong et al., 2018), and reinforcement learning (Parisotto et al., 2020), owing to their self-attention mechanism's capacity for modeling long-range dependencies. However, their quadratic cost in time and memory hinders direct application to high-dimensional tensors, limiting their use in domains like video analysis, high-dimensional forecasting, and 3D imaging. Several strategies attempt to adapt Transformers for multiway data. Flattening tensors into sequences allows standard architectures (Dosovitskiy et al., 2020), but sacrifices structural information and local dependencies, making positional encodings insufficient. Axial attention (Ho et al., 2019) and spatiotemporal attention (Song et al., 2016) mitigate computational costs by processing one mode at a time but may fail to capture full cross-dimensional interactions.

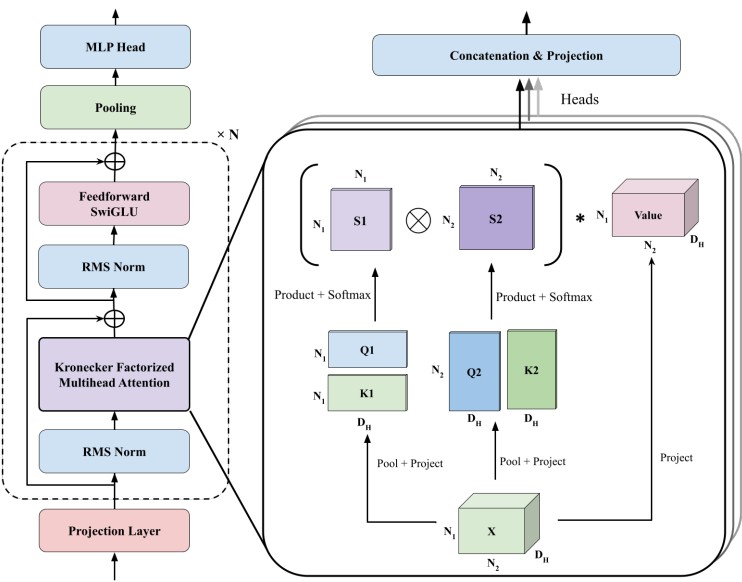

Figure 1: Overall structure of High Order Transformer (HOT) depicting the proposed method for 2D data with size $N_1 \times N_2 \times D$. The model shares the same arrangement as the Transformer encoder while employing Kronecker Factorized Multihead Attention to reduce the computational complexity. Each mode of the tensor (e.g., $N_1, N_2$) has its own attention matrix, combined using Kronecker product operations.

We introduce **Higher-Order Transformers (HOT)**, a novel architecture for efficiently modeling tensor-structured data. Inspired by Kronecker-based covariance models in statistics (Song et al., 2023; Wang et al., 2022), HOT employs a factorized high-order attention mechanism that models both dense and sparse interactions via Kronecker-structured formulations. Each attention head computes independent attention matrices for each modality, which are then combined through Kronecker products or sums. This design preserves the expressivity of full attention while significantly reducing computational costs. Our key contributions are as follows:

- We introduce Higher-Order Transformers (HOT), a factorized attention architecture that extends self-attention to tensor inputs, efficiently capturing dense and sparse cross-dimensional dependencies while preserving the tensor structure.

- We provide theoretical analysis showing that Kronecker factorized attention retains the expressiveness of full high-order attention with controllable computational complexity via factorization rank.

- We validate HOT across diverse benchmarks—including multivariate time series forecasting, 3D medical image classification, and multispectral segmentation—demonstrating competitive or superior performance with significantly reduced computation and memory usage.

## 2 Related Work

Traditional multiway data analysis relies on tensor decompositions such as CP and Tucker to extract latent structures (Kolda & Bader, 2009b). While effective for linear interactions, these methods struggle with complex dependencies. Kronecker structures have been applied to covariance modeling, enabling large matrices to be represented as products of smaller ones (Tsiligkaridis & Hero, 2013; Greenewald & Hero, 2015). Recent advances include iterative algorithms for approximating separable structures (Song et al., 2023) and sparse corrections for improved robustness (Greenewald & Hero, 2015).

Deep learning has driven significant progress in multiway data analysis through CNNs and hierarchical models (LeCun et al., 2015). However, such networks often require millions of parameters. Tensor decompositions have been used to compress network weights while maintaining performance (Novikov et al., 2015; Lebedev et al., 2015). Transformers (Vaswani et al., 2017) excel in sequence modeling but face challenges with high-dimensional data due to computational costs.

A common strategy flattens multi-dimensional data into sequences for Transformer inputs. Vision Transformers (ViT) divide images into patches as tokens (Dosovitskiy et al., 2020), while ViViT extends this to spatiotemporal video patches (Arnab et al., 2021). However, flattening disrupts intrinsic multiway relationships. To preserve structure, tensor decomposition techniques compress Transformer models (Ma et al., 2019), and Kronecker Attention Networks (KAN) use Kronecker operators to capture second-order covariances without flattening (Gao et al., 2020), though with strong distributional assumptions. TimeSFormer (Bertasius et al., 2021) introduces a factorized attention mechanism for video understanding that separately applies attention spatial and temporal dimensions in order. Multiscale Vision Transformers (MViT) (Fan et al., 2021) further improve efficiency by progressively reducing spatial resolution while expanding channel capacity, enabling scalable representation learning across different granularities.

New models explicitly capture cross-dimensional dependencies. iTransformer enhances multivariate time series forecasting by applying attention across variables (Liu et al., 2024), while Crossformer targets spatial-temporal relationships (Zhang & Yan, 2023). In medical imaging, SegFormer3D employs hierarchical cross-scale attention for 3D segmentation (Perera et al., 2024), and CdTransformer distills attention across orthogonal planes to capture anatomical dependencies (Zhu et al., 2024).

The computational demands of self-attention have inspired efficiency-focused innovations. Tensor Product Attention (TPA) compresses queries, keys, and values via tensor decompositions (Zhang et al., 2025). Sparse attention (Child et al., 2019; Zaheer et al., 2020) and linearized methods (Katharopoulos et al., 2020; Choromanski et al., 2021) reduce complexity for longer sequences. Performers approximate attention via kernel methods (Choromanski et al., 2021), while Reformer uses locality-sensitive hashing for efficient memory use (Kitaev et al., 2020).

## 3 Preliminaries

In this section, we introduce key tensor notations and operations that are fundamental to the high-order attention mechanism proposed in this work. Vectors are denoted by lowercase letters (e.g., $v$), matrices by uppercase letters (e.g., $M$), and tensors by calligraphic letters (e.g., $\mathcal{T}$). We use $I_d$ to denote the $d \times d$ identity matrix and $\times_i$ to denote the tensor product along mode $i$ (Kolda & Bader, 2009a). The notation $[k]$ refers to the set $\{1, 2, \ldots, k\}$ for any integer $k$.

**Definition 3.1** (Tensor). A $k$-th order tensor $\mathcal{T} \in \mathbb{R}^{N_1 \times N_2 \times \cdots \times N_k}$ generalizes the concept of a matrix to higher dimensions. A tensor can be viewed as a multidimensional array, where each element is indexed by $k$ distinct indices, representing multiway data that spans across $k$ multiple modes. The number of modes is equal to the order of the tensor.

**Definition 3.2** (Tensor Mode and Fibers). A mode-$i$ fiber of a tensor $\mathcal{T}$ is the vector obtained by fixing all indices of $\mathcal{T}$ except the $i$-th one, e.g., $\mathcal{T}_{n_1,n_2,\dots,n_{i-1},:,n_{i+1},\dots,n_k} \in \mathbb{R}^{N_i}$.

**Definition 3.3** (Tensor Slice). A tensor slice is a two-dimensional section of a tensor, obtained by fixing all but two indices, e.g., $\mathcal{T}_{n_1,n_2,\dots,n_{i-1},:,n_{i+1},\dots,n_{j-1},:,n_{j+1},\dots,n_k} \in \mathbb{R}^{N_i \times N_j}$

Slices and fibers extend the familiar concept of matrix rows and columns to higher-dimensional tensors, providing powerful ways to analyze and manipulate multi-way data.

**Definition 3.4** (Tensor Matricization). The $i$-th mode matricization of a tensor rearranges the mode-$i$ fibers of the tensor into a matrix. It is denoted as $\mathcal{T}_{(i)} \in \mathbb{R}^{N_i \times (N_1 \cdots N_{i-1} N_{i+1} \cdots N_k)}$.

**Definition 3.5** (Mode $n$ tensor product). The mode $n$ product between a tensor $\mathcal{T} \in \mathbb{R}^{N_1 \times N_2 \times \cdots \times N_k}$ and a matrix $A \in \mathbb{R}^{d \times N_n}$ is denoted by $\mathcal{T} \times_n A \in \mathbb{R}^{N_1 \times N_2 \times \cdots \times N_{n-1} \times d \times N_{n+1} \times \cdots \times N_k}$ and defined by $(\mathcal{T} \times_n A)_{i_1,\cdots,i_k} = \sum_j \mathcal{T}_{i_1,\cdots,i_{n-1},j,i_{n+1},\cdots,i_k} A_{i_n,j}$ for all $i_1 \in [N_1], \cdots, i_k \in [N_k]$.

**Definition 3.6** (Rank-1 Tensor). An order-$N$ tensor is of rank-1 if it can be strictly decomposed into the outer product of N vectors. A rank-1 matrix can therefore be written as $X = a \odot b = ab^T$ and a rank-1 order-3 tensor as $\mathcal{X} = a \odot b \odot c$ which can be extended to any-order tensors.

We conclude this part by defining the Kronecker product and sum and stating a useful identity relating matricization, mode $n$ product, and the Kronecker product.

**Definition 3.7** (Kronecker Product and Sum). The Kronecker product of two matrices $A \in \mathbb{R}^{m \times n}$ and $B \in \mathbb{R}^{p \times q}$ is the $mp \times nq$ block matrix

$$A \otimes B = \begin{bmatrix} a_{11}B & \cdots & a_{1,n}B \\ \vdots & \ddots & \vdots \\ a_{m1}B & \cdots & a_{m,n}B \end{bmatrix}.$$

The Kronecker sum of two square matrices $A \in \mathbb{R}^{m \times m}$ and $B \in \mathbb{R}^{n \times n}$ is the $mn \times mn$ square matrix defined by $A \oplus B = A \otimes I_n + I_m \otimes B$.

**Proposition 3.8.** *For any tensor $\mathcal{T} \in \mathbb{R}^{N_1 \times N_2 \times \cdots \times N_k \times d}$ of order $k+1$ and any matrices $A_1 \in \mathbb{R}^{M_1 \times N_1}, \cdots, A_k \in \mathbb{R}^{M_k \times N_k}$, we have $(\mathcal{T} \times_1 A_1 \times_2 A_2 \times_3 \cdots \times_k A_k)_{(k+1)} = \mathcal{T}_{(k+1)}(A_1 \otimes A_2 \otimes \cdots \otimes A_k)^\top$.*

# 4 Method

In this section, we first review the self-attention mechanism in Transformer layers (Vaswani et al., 2017), which we extend to higher orders by tensorizing queries, keys, and values. Then, we explain the theoretical backbone of the proposed low-rank factorized attention and architecture details.

## 4.1 Generalizing Attention to Higher Order Tensors

Given an input tensor $\mathcal{X} \in \mathbb{R}^{N_1 \times N_2 \times \cdots \times N_k \times D}$, where $N_1, N_2, \dots, N_k$ are the sizes of the positional modes and $D$ is the hidden dimension, we start by generalizing the attention mechanism to operate over all positional modes collectively.

The simplest and most naive approach is to flatten the multidimensional input into a sequence, apply standard attention on the resulting matrix, and then reshape it back to the original dimensions. We first compute the query ($\mathcal{Q}$), key ($\mathcal{K}$), and value ($\mathcal{V}$) tensors for each head $h$ by linear projections along the hidden dimension:

$$\mathcal{Q}^h = \mathcal{X} \times_{k+1} (W_Q^h)^\top \in \mathbb{R}^{N_1 \times \cdots \times N_k \times D_H},$$
$$\mathcal{K}^h = \mathcal{X} \times_{k+1} (W_K^h)^\top \in \mathbb{R}^{N_1 \times \cdots \times N_k \times D_H},$$
$$\mathcal{V}^h = \mathcal{X} \times_{k+1} (W_V^h)^\top \in \mathbb{R}^{N_1 \times \cdots \times N_k \times D_H}$$

where $\times_{k+1}$ denotes multiplication along the $(k+1)$-th mode (the hidden dimension).

The scaled dot-product attention scores $S^h \in \mathbb{R}^{(N_1 N_2 \dots N_k) \times (N_1 N_2 \dots N_k)}$ are then given by

$$S^h = \text{Softmax} \left( \frac{(\mathcal{Q}_{(k+1)}^h)^\top \mathcal{K}_{(k+1)}^h}{\sqrt{D_H}} \right) \tag{1}$$

where $\mathcal{Q}_{(k+1)}^h \in \mathbb{R}^{(N_1 N_2 \dots N_k) \times D_H}$ and $\mathcal{K}_{(k+1)} \in \mathbb{R}^{(N_1 N_2 \dots N_k) \times D_H}$ are the matricization of the query and key tensors, and the Softmax function is applied row-wise. Each positional index is considered as a single entity in the attention calculation. The output of the high-order attention function $h_{\text{Attn}} : \mathbb{R}^{N_1 \times N_2 \times \dots \times N_k \times D} \to \mathbb{R}^{N_1 \times N_2 \times \dots \times N_k \times D}$ is computed by applying the attention weights to the value tensor:

$$h_{\text{Attn}}(\mathcal{X})_{(k+1)} = \sum_h (W_O^h)^\top \mathcal{V}_{(k+1)}^h S^h. \tag{2}$$

Lastly, the output is reshaped back to the original tensor shape $N_1 \times N_2 \times \dots \times N_k \times D$.

Although scaled dot-product attention is widely used and has shown great promise across various domains, it comes with limitations that highly impact its scalability. While the above formulation leads to a computational and memory complexity of $\mathcal{O}(D(N_1 N_2 \dots N_k)^2)$, impractical for large tensors, it is also originally designed for 1D sequences and can not directly handle multiway data (e.g., images, videos, etc.) without destroying the tensor structure through flattening. These limitations motivate the development of high-order attention mechanisms that can naturally handle high-order tensor data in an efficient manner.

## 4.2 Kronecker Decomposition

We parameterize the potentially large high-order attention matrix $S^h \in \mathbb{R}^{(N_1 N_2 \dots N_k) \times (N_1 N_2 \dots N_k)}$ using a Kronecker product or sum structure (Figure 2) of the form:

$$S_{prod}^h = S_h^{(1)} \otimes S_h^{(2)} \otimes \dots \otimes S_h^{(k)} \tag{3}$$

$$S_{sum}^h = \frac{1}{k}(S_h^{(1)} \oplus S_h^{(2)} \oplus \dots \oplus S_h^{(k)}) \tag{4}$$

where each $S_h^{(i)} \in \mathbb{R}^{N_i \times N_i}$ is a factor matrix corresponding to the attention weights over the $i$-th mode for head $h$, and $\frac{1}{k}$ is a scalar normalization to ensure the row-stochastic property. Based on these formulations, we propose two variants of our method, namely HOT (product) and HOT (sum).

**Theorem 4.1** (Row stochastic property). *Let $S \in \mathbb{R}^{N \times N}$ and $T \in \mathbb{R}^{M \times M}$ be row-stochastic matrices (i.e. $\sum_j S_{ij} = 1$ for all $i$, and likewise $\sum_\ell T_{k\ell} = 1$ for all $k$). Then $S \otimes T \in \mathbb{R}^{NM \times NM}$ and $\frac{1}{2}(S \oplus T) \in \mathbb{R}^{NM \times NM}$ are also row-stochastic.*

*Proof.* Proof is presented in the appendix A.1. $\square$

**Corollary 4.2.** *Let $\{S^{(i)} \in \mathbb{R}^{N_i \times N_i}\}_{i=1}^k$ be a collection of row-stochastic matrices. Then $S^{(1)} \otimes S^{(2)} \otimes \dots \otimes S^{(k)}$ and $\frac{1}{k}\left(S^{(1)} \oplus S^{(2)} \oplus \dots \oplus S^{(k)}\right)$ are also row-stochastic.*

*Proof.* Proof is presented in the appendix A.1. $\square$

Now we delve into the computation of the factor matrices $S^{(i)}$. As mentioned before, each matrix $S_h^{(i)}$ represents first-order attention weights over the mode $i$. Thus, they can be computed independently using the standard scaled dot-product attention mechanism. Since the input to the attention module is a high-order tensor, computing first-order attention matrices requires reshaping of the input query, key, and value tensors. We propose to use a permutation-invariant pooling functions $g_{\text{pool}}^{(i)} : \mathbb{R}^{N_1 \times \dots \times N_k \times D_H} \to \mathbb{R}^{N_i \times D_H}$ that takes a high-order tensor as input and only preserves the $i$-th mode and the hidden dimension. In this work, we consider summation over all modes except the $i$-th and last one as the pooling function, i.e.,

$$\left(g_{\text{pool}}^{(i)}(\mathcal{T})\right)_{j,l} = \sum_{j_1, \dots, j_{i-1}, j_{i+1}, \dots, j_k} \mathcal{T}_{j_1, \dots, j_{i-1}, j, j_{i+1}, \dots, j_k, l}.$$

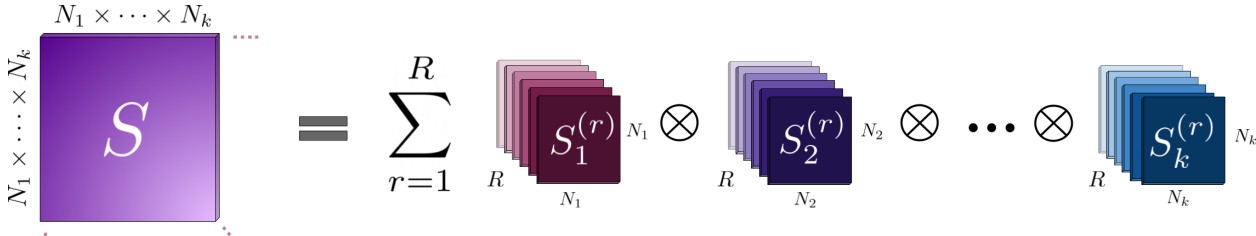

Figure 2: Visualization of a rank $R$ Kronecker Decomposition of a high-order full attention matrix $S \in \mathbb{R}^{N_1 N_2 ... N_k \times N_1 N_2 ... N_k}$ with factor matrices $S_i \in \mathbb{R}^{N_i \times N_i}$. Note that the actual full attention matrix on the LHS can be potentially much larger than what is depicted in the figure.

We then construct the query and key matrices $\tilde{Q}_i^h, \tilde{K}_i^h \in \mathbb{R}^{N_i \times D_H}$ for mode $i$ and head $h$ using weights $W_{query}^{i,h}, W_{key}^{i,h} \in \mathbb{R}^{D_H \times D_H}$ as:

$$\tilde{Q}_i^h = g_{\text{pool}}^{(i)}(\mathcal{X})W_{query}^{i,h} \tag{5}$$

$$\tilde{K}_i^h = g_{\text{pool}}^{(i)}(\mathcal{X})W_{key}^{i,h} \tag{6}$$

Then compute the attention matrix $S_h^{(i)}$ for the $i$th modality as:

$$S_h^{(i)} = \text{Softmax}\left(\frac{\tilde{Q}_i^h(\tilde{K}_i^h)^\top}{\sqrt{D_H}}\right) \tag{7}$$

at a computational cost of $\mathcal{O}(N_i^2 D_H + D_H \prod_j N_j)$.

Explicitly constructing the full attention matrix $S^h$ using the Kronecker product (Eq. equation 3) or sum (Eq. equation 4) formulation from the factor matrices $S_h^{(i)}$ would negate the computational savings of the factorization. Instead, we exploit properties of the Kronecker product and the associative law of matrix and tensor multiplication to apply the attention without forming $S^h$ directly. Formally, it is easy to check that

$$S_{prod}^h(\mathcal{V}_{(k+1)}^h)^\top = \left(S_h^{(1)} \otimes ... \otimes S_h^{(k)}\right)(\mathcal{V}_{(k+1)}^h)^\top = \left(\left(\mathcal{V}^h \times_1 S_h^{(1)}\right) \times_2 ... \times_k S_h^{(k)}\right)_{(k+1)}^\top \tag{8}$$

and

$$S_{sum}^h(\mathcal{V}_{(k+1)}^h)^\top = \frac{1}{k}\left(S_h^{(1)} \oplus ... \oplus S_h^{(k)}\right)(\mathcal{V}_{(k+1)}^h)^\top = \frac{1}{k}\left(\left(\mathcal{V}^h \times_1 S_h^{(1)}\right) + ... + \left(\mathcal{V}^h \times_k S_h^{(k)}\right)\right)_{(k+1)}^\top \tag{9}$$

We can, thus, multiply attention matrices one by one with the value tensor. The operation on each mode $i$ yields a computational complexity of $\mathcal{O}(N_i D_H(\prod_j N_j))$, resulting in an overall complexity of $\mathcal{O}(D_H(\sum_i N_i)(\prod_j N_j))$. For a factorized attention layer with $R$ heads of width $D_H = D/R$, this leads to a total complexity is $\mathcal{O}(D(\sum_i N_i)(\prod_j N_j))$.

In addition to reducing computational and memory complexity, the proposed factorization of the high-order attention matrix provides a convenient framework for studying and customizing attention on each mode independently. This not only simplifies post-training interpretation of attention maps along each axis but also enables the use of custom attention masks tailored to specific modalities and applications. For instance, a causal mask can be easily applied to temporal attention in an autoregressive generative setting. However, in this paper, we focus solely on the use case of HOT as a Transformer encoder.

### 4.3 Architecture

Our final architecture is built on three main components (Fig. 1):

1. A projection layer implemented as a convolution layer followed by a ReLU activation function that projects the input tensor to the hidden space. Setting the stride equal to the kernel size, this module divides the input into non-overlapping temporal or spatial patches, reducing the input data resolution.

2. The Transformer encoder layer consists of alternating layers of our proposed Kronecker (product or sum) self-attention and MLP blocks. Layernorm is applied before every block, and residual connections after every block. The MLP contains two layers with a GeLU non-linearity.

3. A global average pooling layer, followed by a single linear layer as the projection head.

## 4.4 Theoretical Analysis

### 4.4.1 Attention Matrix Rank

Understanding the rank of attention matrices is key to analyzing Transformer expressivity and efficiency. Although we model high-order attention via low-rank factorization, this does not imply that the computed matrices $S_{prod}^h$ and $S_{sum}^h$ from Eqs. 3 and 4 are themselves low-rank. Instead, we analyze their geometry and spectral spread as proxies for effective rank.

While attention matrices are theoretically full-rank, since softmax of random matrices typically yields linearly independent rows, their effective rank is often much lower due to row-stochasticity, dominance of a few singular values from peaked attention distributions, and training dynamics encouraging low-rank structures. Moreover, computing the exact rank of an $n \times n$ attention matrix is computationally expensive at $\mathcal{O}(n^3)$ and numerically unstable. Therefore, we adopt the *stable rank* as a robust measure, defined for any nonzero $A \in \mathbb{R}^{m \times n}$ and its singular values $s_i(A)$ as:

$$\mathrm{sr}(A) = \frac{|A|_F^2}{|A|_2^2} = \frac{\sum_i s_i^2(A)}{s_1^2(A)}. \tag{10}$$

We analyze stable rank for the factorized attention matrices $S_{\mathrm{prod}}^h$ and $S_{\mathrm{sum}}^h$, as well as the mode-wise matrices $S_h^{(i)}$. For the Kronecker product, the stable rank factorizes:

$$\mathrm{sr}(S_{\mathrm{prod}}^h) = \mathrm{sr}(S_h^{(1)}) \times \cdots \times \mathrm{sr}(S_h^{(k)}). \tag{11}$$

For the Kronecker sum formulation, the stable rank is

$$\mathrm{sr}(S_{\mathrm{sum}}^h) = \frac{\sum_{i=1}^k \left(\prod_{j \neq i} n_j\right)|S_h^{(i)}|_F^2 + \sum_{i,l}\left(\prod_{j \neq i,l} n_j\right)\mathrm{tr}(S_h^{(i)})\,\mathrm{tr}(S_h^{(l)})}{\left(\sum_{i=1}^k |S_h^{(i)}|_2\right)^2}. \tag{12}$$

Before deriving bounds, we recall for any square $A \in \mathbb{R}^{n \times n}$:

$$1 \leq |A|_F^2 \leq n, \tag{13}$$
$$1 \leq \mathrm{tr}(A) \leq n, \tag{14}$$
$$0 \leq s_i(A) \leq 1, \quad 0 < s_1(A) \leq 1, \tag{15}$$

implying

$$1 \leq \mathrm{sr}(A) \leq n. \tag{16}$$

Thus, the stable ranks satisfy

$$1 \leq \mathrm{sr}(S_{\mathrm{prod}}^h) \leq \prod_{i=1}^k N_i, \tag{17}$$

and

$$\frac{\sum_{i=1}^k \left(\prod_{j \neq i} N_j\right) + \sum_{i,j=1}^k \left(\prod_{l \neq i,j} N_l\right)}{k^2} \leq \mathrm{sr}(S_{\mathrm{sum}}^h) \leq \prod_{i=1}^k N_i. \tag{18}$$

While $\mathrm{sr}(S_{\mathrm{prod}}^h)$ shares the same bounds as the original non-factorized attention matrix, $\mathrm{sr}(S_{\mathrm{sum}}^h)$ has a higher lower bound, independent of the stable ranks of the mode-wise matrices $S_h^{(i)}$. This means that even if all $S_h^{(i)}$ collapse to rank-one, $S_{\mathrm{sum}}^h$ typically avoids collapsing, especially when the $N_i$ are large relative to $k$. In the next chapter, we empirically compare stable ranks for the Kronecker product and sum attention matrices.

### 4.4.2 Low Rank Factorization Approximation Guarantee

The summation across all heads appearing in Eq. equation 2 when using Kronecker product formulation from Eq. 3 functions analogously to a rank $R$ Kronecker decomposition (when $W_O^h$ is the same for all heads) and the factorization rank corresponds to the number of heads. Note that the *factorization rank* is different and independent from the *attention matrix rank* (explained in the previous section) as the former is a hyperparameter and the latter is to be computed in practice. In Theorem 4.3 below we show that any attention matrix can be decomposed as a sum of such Kronecker products. The following theorem shows that a rank $R$ Kronecker decomposition is capable of approximating any high-order attention matrix arbitrarily well, as $R$ increases, ensuring that no significant interactions are missed. This theoretical aspect is crucial for ensuring that the attention mechanism can potentially adapt to any dataset or task requirements.

**Theorem 4.3** (Universality of Kronecker product decomposition). *Given any high-order attention matrix $S \in \mathbb{R}^{(N_1 N_2 ... N_k) \times (N_1 N_2 ... N_k)}$, there exists an $R \in \mathbb{N}$ such that $S$ can be expressed as a rank $R$ Kronecker decomposition, i.e., $S = \sum_{r=1}^{R} S_r^{(1)} \otimes S_r^{(2)} \otimes ... \otimes S_r^{(k)}$. As $R$ approaches $\min_{j=1,\cdots,k} \prod_{i \neq j} N_i^2$, the approximation is guaranteed to become exact, meaning the Kronecker decomposition is capable of universally representing any high-order attention matrix $S$.*

*Proof.* Proof is presented in the appendix A.1. $\square$

While a sum of Kronecker products can express any tensor (given enough summands), it is not the case for sums of Kronecker sums (which are still Kronecker sums). Still, our experiments showcase that the Kronecker sum decomposition can almost always match the performance of the Kronecker product decomposition, suggesting that the attention matrices needed to obtain good accuracy on multi-modal data are highly structured. In the appendix, we have discussed the choice of Kronecker factorization in section A.2 and the inductive biases of our method in section A.3.

## 5 Experiments

We thoroughly evaluate HOT on three high-order data tasks, validating the generality of the proposed framework. At each subsection, we introduce the task, benchmark datasets, and baselines used, and discuss the performance results. Implementation details are presented in the appendix. We close the section by reviewing ablation studies that further confirm our theory and design choices.

Table 1: Multivariate forecasting results with prediction lengths $S \in \{96, 192, 336, 720\}$ and fixed lookback length $T = 96$ with the best in **Bold** and second-best in underline. Results are averaged from all prediction lengths. FLOPS are calculated for a fixed input of size (100, 96). $^\dagger$Results reported from the original papers; others are reproduced by us.

| Models | Params | GFLOPS | ECL | | Weather | | Traffic | | Solar | |
|---|---|---|---|---|---|---|---|---|---|---|
| | | | MSE | MAE | MSE | MAE | MSE | MAE | MSE | MAE |
| AutoFormer$^\dagger$ | 15M | - | 0.227 | 0.338 | 0.338 | 0.382 | 0.628 | 0.379 | 0.885 | 0.711 |
| FedFormer$^\dagger$ | 21M | - | 0.214 | 0.327 | 0.309 | 0.360 | 0.610 | 0.376 | 0.291 | 0.38 |
| Crossformer$^\dagger$ | - | - | 0.244 | 0.334 | 0.259 | 0.315 | 0.550 | 0.304 | 0.641 | 0.639 |
| TimesNet$^\dagger$ | 301M | - | 0.192 | 0.295 | 0.259 | 0.287 | 0.620 | 0.336 | 0.301 | 0.319 |
| PatchTST$^\dagger$ | 1.5M | - | 0.205 | 0.290 | 0.259 | 0.281 | 0.481 | 0.304 | 0.270 | 0.307 |
| Transformer (spatial)$^\dagger$ | 13M | 1.03 | 0.178 | 0.270 | 0.258 | 0.278 | 0.428 | 0.282 | 0.233 | 0.262 |
| Transformer (temporal)$^\dagger$ | - | 1.00 | 0.202 | 0.300 | 0.258 | 0.283 | 0.863 | 0.499 | 0.285 | 0.317 |
| Transformer (non-factorized) | 620K | 4.22 | **0.162** | **0.261** | **0.245** | **0.275** | OOM | OOM | **0.219** | **0.257** |
| HOT (product) | 680K | 1.51 | 0.169 | 0.268 | **0.245** | **0.275** | **0.420** | **0.278** | 0.221 | **0.257** |
| HOT (sum) | 680K | 1.51 | 0.167 | 0.266 | **0.245** | **0.275** | 0.428 | 0.280 | 0.220 | 0.260 |

## 5.1 Long-range Time-series Forecasting

Given historical observations $X = \{\mathbf{x}_1, \ldots, \mathbf{x}_T\} \in \mathbb{R}^{T \times N}$ with $T$ time steps and $N$ variates, we predict the future $S$ time steps $Y = \{\mathbf{x}_{T+1}, \ldots, \mathbf{x}_{T+S}\} \in \mathbb{R}^{S \times N}$.

**Datasets** We include four real-world datasets in our experiments, including ECL, Traffic, Weather used by Autoformer (Wu et al., 2021), and Solar-Energy proposed in LSTNet (Lai et al., 2017). Further dataset details are in the Appendix.

**Baselines** We choose seven Transformer-based models as our baselines, including Crossformer (Zhang & Yan, 2023), Autoformer (Wu et al., 2021), FEDformer (Zhou et al., 2022), PatchTST (Nie et al., 2023), and three variations of the vanilla Transformer based on how it processes timeseries data: 1. Transformer (spatial) with attention applied on spatial axis (Liu et al., 2024) 2. Transformer (temporal) with attention applied on the temporal axis (Liu et al., 2024) 3. Transformer (non-factorized) with full spatiotemporal attention on flattened input corresponding to non-factorized high-order attention.

**Results** Table 1 presents the forecasting results, with the best and second-best performances highlighted. The non-factorized Transformer achieves the best results across all datasets except Traffic, where it runs out of memory. While this demonstrates its strong expressive power, it comes at a significant computational cost due to the quadratic complexity of attention with respect to the product of data dimensions. Our proposed HOT models offer a more efficient alternative. HOT (product) matches the performance of the non-factorized Transformer on most datasets while requiring only a third of the floating-point operations. HOT (sum) consistently ranks second, closely following HOT (product). Both models also use significantly fewer parameters, making them computationally efficient. Among the baselines, the spatial Transformer (i.e., iTransformer (Liu et al., 2024)) outperforms the temporal Transformer and several methods like Autoformer, FedFormer, and PatchTST, showcasing the importance of capturing spatial correlations in high-dimensional data.

Table 2: 3D image classification results on MedMNIST3D with the best in **Bold** and second-best in underline. FLOPS are calculated for a single input image. [†]Results reported from the original papers; others are reproduced by us.

| Models | Params | GFLOPS | Organ | | Nodule | | Fracture | | Adrenal | | Vessel | |
|---|---|---|---|---|---|---|---|---|---|---|---|---|
| | | | AUC | ACC | AUC | ACC | AUC | ACC | AUC | ACC | AUC | ACC |
| ResNet-18[†] | 12M | 2.32 | 99.4 | 90.0 | 86.3 | 84.4 | 71.2 | 50.8 | 82.7 | 72.1 | 82.0 | 74.5 |
| ResNet-50[†] | 26M | 3.19 | 99.4 | 88.9 | 88.6 | 84.1 | **75.0** | 51.7 | 82.8 | 75.8 | 91.2 | 85.8 |
| MDANet[†] | 7M | - | 98.9 | 89.7 | 86.8 | 86.00 | - | - | 83.9 | 81.5 | 90.1 | **92.9** |
| CdTransformer[†] | - | - | - | - | 91.9 | 88.6 | 71.6 | 51.7 | 87.7 | 81.5 | 91.9 | 89.3 |
| ViT-3D[†] | 130M | - | - | - | 91.4 | 86.7 | 64.8 | 53.3 | 82.0 | 81.9 | 82.6 | 90.1 |
| ViViT-S[†] | 50M | - | - | - | 86.6 | 85.8 | 65.5 | 53.8 | 81.0 | 79.9 | 83.9 | 88.7 |
| TimeSformer | 2M | 0.71 | 99.4 | 90.6 | 89.8 | 86.4 | 68.0 | 52.3 | 81.7 | 82.5 | 85.3 | 91.3 |
| MViT | 1.6M | 0.72 | 99.0 | 88.1 | 90.1 | 86.8 | 66.7 | 53.7 | 82.5 | 80.1 | 80.6 | 89.2 |
| Transformer (non-factorized) | 1.5M | 0.72 | 99.7 | 92.6 | 91.6 | 87.8 | 74.4 | 58.6 | **88.8** | **83.9** | **92.0** | 91.1 |
| HOT (product) | 2M | 0.67 | **99.8** | **94.4** | **92.0** | **89.1** | 73.6 | 58.3 | 88.6 | 83.8 | 85.7 | 91.1 |
| HOT (sum) | 2M | 0.67 | 99.6 | 92.2 | 90.7 | 87.7 | 74.9 | **58.7** | 87.3 | 82.9 | 84.9 | 90.7 |

## 5.2 3D Medical Image Classification

Given a 3D image $\mathcal{X} \in \mathbb{R}^{W \times H \times D}$ with width $W$, height $H$, and depth $D$, we predict the image class probability $y \in \mathbb{R}^C$ over a set of $C$ classes.

**Dataset** MedMNIST v2 (Yang et al., 2023) is a large-scale benchmark for medical image classification on standardized MNIST-like 2D and 3D images with diverse modalities, dataset scales, and tasks. We primarily experiment on the 3D portion of MedMNIST v2, namely the Organ, Nodule, Fracture, Adrenal, and Vessel datasets. The size of each image is $28 \times 28 \times 28$ (3D).

**Baselines**   We choose seven medical image classifier models, including ResNet-18/ResNet-50 (He et al., 2015; Yang et al., 2023), MDANet (Huang et al., 2022), CdTransformer (Zhu et al., 2024), ViT-3D and ViViT-S (Lai et al., 2024), TimeSFormer (Bertasius et al., 2021), MViT (Fan et al., 2021), and vanilla Transformer with full attention over all three axes applied on flattened input as our baselines.

**Results**   The classification results on MedMNIST3D are summarized in Table 2, with the best and second-best performances highlighted. HOT (product) achieves state-of-the-art results on most datasets, with the highest AUC and accuracy for Organ, Nodule, and Adrenal datasets, while requiring minimal computational resources. HOT (sum) closely follows, demonstrating competitive performance, particularly on the Fracture dataset, where it achieves the best accuracy. Both HOT variants utilize only 2M parameters and maintain a low FLOPS count, showcasing their efficiency. The full Transformer achieves strong results, often ranking second or matching HOT (product), especially on Adrenal and Vessel datasets. Among other transformer baselines, TimeSFormer and MViT also deliver competitive results with very low FLOPS, with TimeSFormer reaching second-best AUC on Vessel and MViT showing balanced performance across Nodule and Fracture, though both remain behind HOT overall. Additionally, ResNet-50 performs well on certain datasets, particularly Fracture, but lags in computational efficiency. CdTransformer and ViT-3D demonstrate strong performance on specific datasets (e.g., Nodule and Vessel for CdTransformer), but their parameter-heavy designs make them less efficient compared to HOT. Overall, HOT models strike the best balance between accuracy and efficiency across diverse 3D image classification tasks.

Table 3: Multispectral image segmentation results on SSL4EO-L Benchmark dataset (using OLI/TIRS-RS sensor product) with the best in **Bold** and second-best in underline. [†]Results reported from the original papers; others are reproduced by us.

| Models | Pretraining | Params | GFLOPS | CDL | NLCD |
|---|---|---|---|---|---|
| ResNet18[†] | | 11.2M | 3.42 | **68.0** | 67.0 |
| ResNet50[†] | MoCo | 23.5M | 7.12 | 65.9 | 67.4 |
| ViT-S16[†] | | 22.5M | 5.07 | 64.1 | 66.8 |
| HOT (product) | - | 4.3M | 3.89 | 66.2 | **68.6** |
| HOT (sum) | | 4.3M | 3.89 | 64.5 | 67.2 |

## 5.3   Multi-Spectral Image Segmentation

Given a multisepctral satellite image $\mathcal{X} \in \mathbb{R}^{W \times H \times S}$ with width $W$, height $H$, and spectral channels $S$, we predict the pixel-wise class probability $y \in \mathbb{R}^{W \times H \times C}$ over a set of $C$ classes.

**Dataset**   The SSL4EO-L Benchmark dataset (Stewart et al., 2023) is a collection of Landsat images paired with land cover classification masks. The images are captured using different sensors; for this experiment, we focus only on images from the OLI/TIRS-RS sensor product. The dataset is available in two versions: NLCD with 17 classes and CDL with 134 classes. The size of each image is $264 \times 264 \times 7$.

**Baselines**   We compare HOT variants with ViT and ResNet results reported in the benchmark paper (Stewart et al., 2023). We only report models pretrained with MoCo (He et al., 2019) and subsequently fine-tuned on the SSL4EO-L Benchmark, as these consistently yield the best performance across all backbones.

**Results**   Table 3 reports multispectral segmentation results on SSL4EO-L. Among MoCo-pretrained baselines, ResNet18 (11.2M params, 3.42 GFLOPS) achieves the best CDL score (68.0), while ResNet50 (23.5M params, 7.12 GFLOPS) attains the second-best NLCD score (67.4), suggesting that convolutional backbones remain strong for high-resolution image segmentation. ViT-S16, despite a similar parameter count (22.5M, 5.07 GFLOPS), underperforms both ResNet variants. Our proposed HOT models, trained from scratch with only 1M parameters and 3.89 GFLOPS, outperform ViT on both benchmarks, achieve the best NLCD result (68.6), and remain competitive with ResNet18 on CDL (66.2), highlighting their efficiency in balancing accuracy and computational cost.

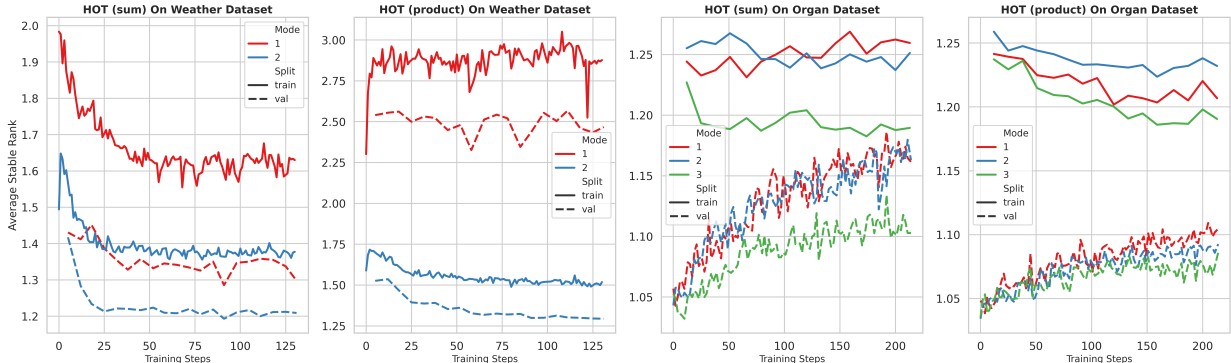

Figure 3: Evolution of the average stable rank for mode-wise attention matrices across training steps for HOT (product) and HOT (sum) models.

## 5.4 Ablation Study

### 5.4.1 Attention Rank

We examine the average stable rank of attention matrices across heads and layers, including mode-wise matrices $S^{(i)} * h$, factorized matrices $S^h * \text{prod}$, $S^h_{\text{sum}}$, and the full attention from the Transformer baseline. On the Weather dataset (Fig.3), HOT (product) shows higher mode-wise stable ranks than HOT (sum), with consistently lower ranks in the time mode compared to the variate mode—despite similar dimensions—suggesting richer spatial dependencies. Stable ranks plateau early and remain steady during training. For the Organ dataset, stable ranks are similar across all three spatial modes in both

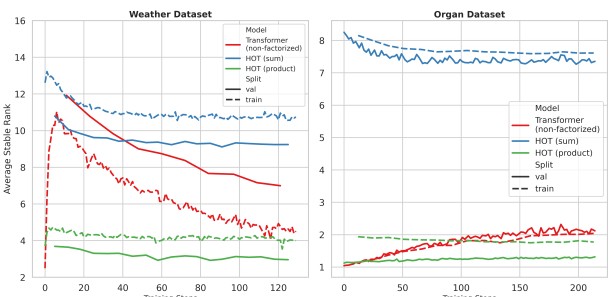

Figure 4: Evolution of the average stable rank for full attention matrices across training steps. Results are shown for the Transformer (non-factorized), HOT (sum), and HOT (product) models.

models, reflecting uniform spatial structure in the $28 \times 28 \times 28$ volumes. Ranks stay stable throughout training, with a slight increase on validation. Training ranks are generally higher due to dropout-induced variability. Looking at full attention matrices (Fig.4), HOT (sum) has consistently higher stable ranks than HOT (product), consistent with theory. Both maintain steady ranks during training. The non-factorized Transformer, however, exhibits an early spike in stable rank before converging near HOT (product) levels—a trend also seen on the Organ dataset and deserving further analysis.

### 5.4.2 Factorization Rank

We analyze how the number of attention heads (i.e., factorization rank) influences HOT's performance on medical imaging and time series datasets. The factorization rank is critical for approximating high-order attention and capturing diverse cross-dimensional patterns. Following common practice, we treat the rank (i.e., number of attention heads) as a hyperparameter and determine the best value through hyperparameter search. Figure 5 shows that increasing the rank generally improves performance, with optimal results between 8 and 16 heads. Beyond this range, higher ranks degrade performance and increase errors, likely due to overfit-

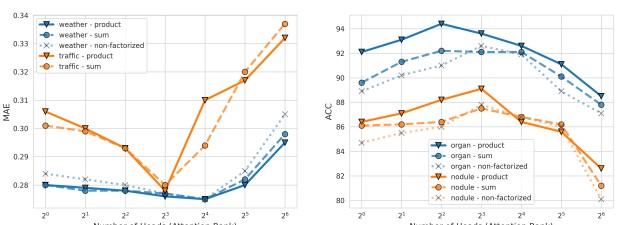

Figure 5: Effect of increasing the number of heads (i.e., factorization rank) on model performance with Kronecker product, Kronecker sum, and non-factorized attention. **Left:** Multivariate time series datasets. **Right:** 3D medical imaging datasets.

ting and reduced head dimension $d_h = D/h$, which eventually becomes too small to model interactions effectively. Optimal ranks vary by dataset: HOT (product) and HOT (sum) perform best at rank 8 on Traffic and Organ, while rank 16 yields superior results on Nodule and Weather.

### 5.4.3 Efficiency Analysis

We evaluate the computational efficiency of HOT against ViT, ResNet50, and ResNet18 across training time, inference time, and GPU memory footprint (Figure 7). HOT requires 27.4 ms per sample for training—higher than ResNet18 (11.0 ms) and ResNet50 (22.0 ms), but lower than ViT (34.2 ms)—and achieves 9.8 ms per sample for inference, placing it between ResNet18 (3.4 ms) and ViT (11.5 ms), while ResNet50 is slightly faster at 7.2 ms. Notably, HOT is the most memory-efficient, using only 0.21 GB of GPU memory, significantly less than ViT (0.74 GB) and even ResNet18 (0.39 GB). We further analyze HOT's Kronecker attention by comparing FLOP requirements against Full (non-factorized) and Divided (from (Bertasius et al., 2021)) attention mechanisms (Figure 6), showing that for an input of size $224^3$, Kronecker attention requires only 371.6 GFLOPS, versus 775.3 GFLOPS for Full attention and 1210 GFLOPS for Divided attention, with the efficiency gap consistent across smaller inputs. Overall, HOT maintains competitive training and inference times, minimizes memory usage, and provides substantial computational savings at scale, making it a strong candidate for resource-constrained medical AI applications. All experiments were implemented in PyTorch and conducted on a single NVIDIA A100 GPU (80 GB) with an x86_64 CPU (6 cores) and 64 GB of RAM.

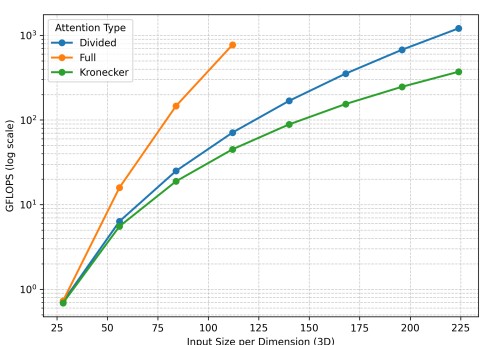

Figure 6: Comparison of computational cost measured in GFLOPS versus 3D input dimension for different attention mechanisms. The Kronecker, Full, and Divided attention types exhibit distinct scaling behaviors as the input dimension increases, with the Kronecker attention having significantly lower computational cost.

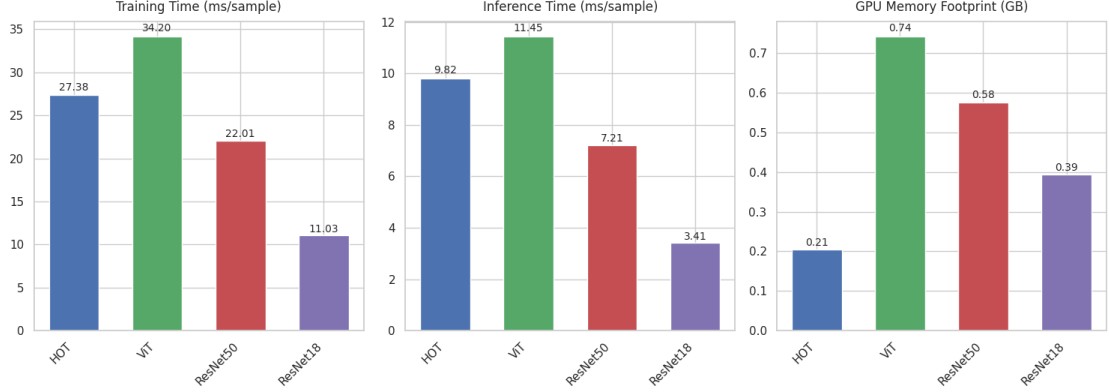

Figure 7: Comparison of model efficiency across four architectures: HOT, ViT, ResNet50, and ResNet18. The figure shows three metrics: (a) training time per sample (ms), (b) inference time per sample (ms), and (c) GPU memory footprint (GB). HOT exhibits the highest training time but low GPU memory usage, while ViT and ResNet variants offer a better trade-off between speed and memory consumption.

### 5.5 Visualization of Attention Maps

We visualize the spatial and temporal attention maps learned by the HOT model on the ECL dataset for time series forecasting. To reveal these patterns, we compute average attention weights across a batch of

test samples for each head and layer. Overall, we observed no significant differences between the patterns learned by the HOT (Product) and HOT (Sum) models. However, clear distinctions emerge between the temporal attention maps and those along the spatial (variable) axis.

Spatial maps (Fig.8) exhibit two main patterns: clustered regions and sparse vertical stripes. The clustered patterns indicate strong dependencies among groups of variables, suggesting the model identifies meaningful partitions in the data. In contrast, the sparse vertical stripes highlight specific variables that exert significant influence on others, reflecting their importance in prediction.

Temporal maps (Fig.9) reveal two characteristic behaviors. First, many attention heads display diagonal line structures, capturing periodic temporal dependencies; the distance of these lines from the main diagonal corresponds to the length of the learned period. Additionally, some attention maps, especially in later layers, become sparse, focusing attention on key time steps such as the beginning or end of the sequence. Complete attention visualizations and additional ablations are provided in the appendix D.

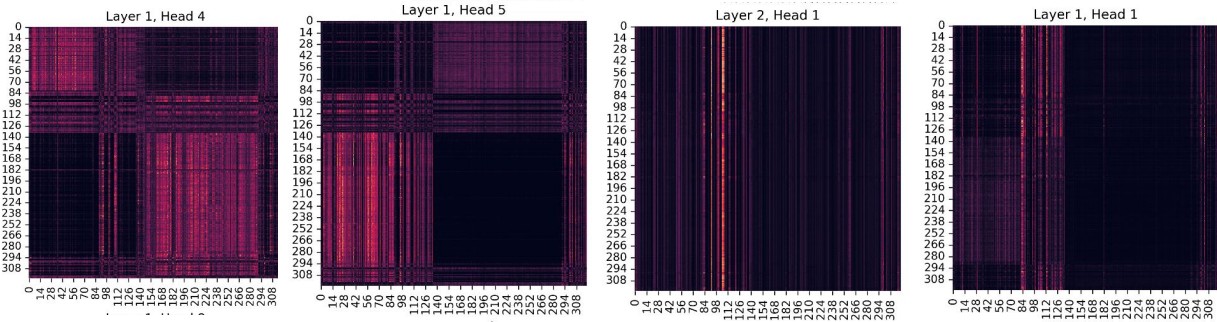

Figure 8: Visualization of the spatial attention maps learned by HOT

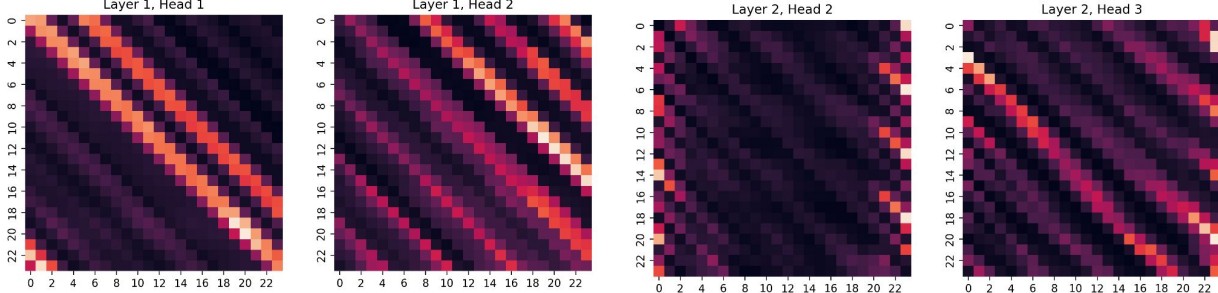

Figure 9: Visualization of the temporal attention maps learned by HOT

## 6 Conclusion

We addressed the challenge of applying Transformers to high-dimensional, multiway tensor data, where standard attention mechanisms are computationally expensive and misaligned with tensor structures. To overcome these limitations, we introduced the Higher-Order Transformer (HOT), which uses Kronecker factorization to express multiway attention as sums of mode-wise attention matrices. This approach efficiently models dependencies across dimensions while preserving tensor structure. Our analysis shows HOT retains the expressiveness of full high-order attention with adjustable complexity through a rank parameter. Experiments on diverse tasks, including multivariate time series forecasting, 3D medical imaging, and multispectral image segmentation demonstrate HOT's strong performance and scalability, while attention visualizations highlight its ability to learn meaningful, interpretable patterns. Future work includes extending HOT to generative modeling, applying it to large-scale image and video datasets, exploring its transfer learning potential across modalities, and integrating it with other tensor decompositions to further enhance efficiency

and modeling capabilities. HOT offers a promising foundation for advancing multiway data modeling across diverse domains.

**Broader impact statement**   This work introduces Higher-Order Transformers (HOT) as an efficient and interpretable framework for modeling multiway data, with experiments on 3D medical imaging benchmarks. Potential benefits include faster and more accurate image analysis, lower computational costs that broaden access, and improved transparency through mode-wise attention maps. However, risks include bias from non-diverse datasets, limited generalization, privacy concerns, and automation bias in clinical use. Since our evaluations are limited to research benchmarks, real-world performance and interpretability remain uncertain. Future work should focus on fairness audits, robust validation on diverse clinical data, uncertainty quantification, and human-in-the-loop deployment to ensure safe and ethical use.

**Acknowledgments**   G. Rabusseau and R. Rabbany acknowledges the support of the CIFAR AI chair program. This work made use of compute resources by the Digital Research Alliance of Canada.

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

## A Theoretical Analysis

### A.1 Proofs

**Theorem A.1** (Row stochastic property). *Let $S \in \mathbb{R}^{N \times N}$ and $T \in \mathbb{R}^{M \times M}$ be row-stochastic matrices (i.e. $\sum_j S_{ij} = 1$ for all $i$, and likewise $\sum_\ell T_{k\ell} = 1$ for all $k$). Then $S \otimes T \in \mathbb{R}^{NM \times NM}$ and $\frac{1}{2}(S \oplus T) \in \mathbb{R}^{NM \times NM}$ are also row-stochastic.*

*Proof.* For the Kronecker product, recall by definition

$$(S \otimes T)_{(i,k),(j,\ell)} = S_{ij}\, T_{k\ell}$$

for $i, j = 1, \ldots, N$ and $k, \ell = 1, \ldots, M$. Fix an output-row index $(i, k)$. Then

$$\sum_{(j,\ell)} (S \otimes T)_{(i,k),(j,\ell)} = \sum_{j=1}^{N}\sum_{\ell=1}^{M} \big(S_{ij}\, T_{k\ell}\big) = \Big(\sum_{j=1}^{N} S_{ij}\Big)\Big(\sum_{\ell=1}^{M} T_{k\ell}\Big) = 1 \times 1 = 1.$$

Since this holds for every row $(i, k)$, $S \otimes T$ is row-stochastic.

For the Kronecker sum, by definition $S \oplus T = S \otimes I_M + I_N \otimes T$. Again, fix a row index $(i, k)$. We have:

$$\sum_{(j,\ell)} \big[(S \otimes I_M)_{(i,k),(j,\ell)} \; + \; (I_N \otimes T)_{(i,k),(j,\ell)}\big] = \sum_{(j,\ell)} (S \otimes I_M)_{(i,k),(j,\ell)} + \sum_{(j,\ell)} (I_N \otimes T)_{(i,k),(j,\ell)}$$

while $(S \otimes I_M)_{(i,k),(j,\ell)} = S_{ij}\, \delta_{k\ell}$, and $(I_N \otimes T)_{(i,k),(j,\ell)} = \delta_{ij}\, T_{k\ell}$. Hence,

$$\sum_{j,\ell} S_{ij}\, \delta_{k\ell} = \Big(\sum_{j=1}^{N} S_{ij}\Big)\Big(\sum_{\ell=1}^{M} \delta_{k\ell}\Big) = 1 \; \times \; 1 = 1,$$

and similarly

$$\sum_{j,\ell} \delta_{ij}\, T_{k\ell} = \Big(\sum_{j=1}^{N} \delta_{ij}\Big)\Big(\sum_{\ell=1}^{M} T_{k\ell}\Big) = 1 \; \times \; 1 = 1.$$

Thus,

$$\sum_{(j,\ell)} \frac{1}{2}(S \oplus T) = \frac{1}{2}(1 + 1) = 1.$$

$\square$

**Corollary A.2.** *Let $\{S^{(i)} \in \mathbb{R}^{N_i \times N_i}\}_{i=1}^{k}$ be a collection of row-stochastic matrices. Then $S^{(1)} \otimes S^{(2)} \otimes \cdots \otimes S^{(k)}$ and $\frac{1}{k}\big(S^{(1)} \oplus S^{(2)} \oplus \cdots \oplus S^{(k)}\big)$ are also row-stochastic.*

*Proof.* The Kronecker product is associative, and from the theorem, we know that the Kronecker product of two row-stochastic matrices is row-stochastic. Now, we proceed by induction. The base case $k = 2$ is given by the theorem. Assume that the Kronecker product of $k-1$ row-stochastic matrices is row-stochastic. Then $\big(S^{(1)} \otimes \cdots \otimes S^{(k-1)}\big) \otimes S^{(k)}$ is the Kronecker product of a row-stochastic matrix with $S^{(k)}$, which by the base theorem is again row-stochastic.

By definition, the Kronecker sum of $k$ matrices is given by a sum of $k$ terms, each of the form:

$$A^{(i)} := I_{N_1} \otimes \cdots \otimes I_{N_{i-1}} \otimes S^{(i)} \otimes I_{N_{i+1}} \otimes \cdots \otimes I_{N_k}.$$

Each term $A^{(i)}$ is a Kronecker product of identity matrices and one row-stochastic matrix $S^{(i)}$, so by the product result above, each $A^{(i)}$ is row-stochastic. Summing $k$ such matrices yields a matrix whose rows sum to $k$, so dividing by $k$ ensures that the normalized sum $\frac{1}{k}\sum_{i=1}^{k} A^{(i)}$ is row-stochastic. $\square$

**Theorem A.3** (Universality of Kronecker product decomposition)**.** *Given any high-order attention matrix* $S \in \mathbb{R}^{(N_1 N_2 \dots N_k) \times (N_1 N_2 \dots N_k)}$*, there exists an* $R \in \mathbb{N}$ *such that* $S$ *can be expressed as a rank* $R$ *Kronecker decomposition, i.e.,* $S = \sum_{r=1}^{R} S_r^{(1)} \otimes S_r^{(2)} \otimes \dots \otimes S_r^{(k)}$*. As* $R$ *approaches* $\min_{j=1,\cdots,k} \prod_{i \neq j} N_i^2$*, the approximation is guaranteed to become exact, meaning the Kronecker decomposition is capable of universally representing any high-order attention matrix* $S$*.*

*Proof.* Let $\mathcal{S}$ be the tensor obtained by reshaping the attention matrix $S$ into a tensor of size $N_1 \times N_2 \times \cdots \times N_k \times N_1 \times N_2 \times \cdots \times N_k$ and let $\mathcal{T} \in \mathbb{R}^{N_1^2 \times N_2^2 \times \cdots \times N_k^2}$ be the tensor obtained by merging each pair of modes corresponding to one modality[1]. Let $R$ be the CP rank of $\mathcal{T}$ and let $\mathcal{T} = \sum_{r=1}^{R} s_r^{(1)} \circ s_r^{(2)} \circ \cdots \circ s_r^{(k)}$ be a CP decomposition, where $\circ$ denotes the outer product and each $s_r^{(i)} \in \mathbb{R}^{N_i^2}$ for $i = 1, \cdots, k$ (see, e.g., (Kolda & Bader, 2009a) for an introduction to the CP decomposition). By reshaping each $s_r^{(i)} \in \mathbb{R}^{N_i^2}$ into a matrix $S_r^{(i)} \in \mathbb{R}^{N_i \times N_i}$, one can check that

$$
\mathcal{S}_{i_1,\cdots,i_k,j_1,\cdots,j_k} = \sum_{r=1}^{R} (S_r^{(1)})_{i_1,j_1} \otimes (S_r^{(2)})_{i_2,j_2} \otimes \cdots \otimes (S_r^{(k)})_{i_k,j_k}
$$

from which it follows that $S = \sum_{r=1}^{R} S_r^{(1)} \otimes S_r^{(2)} \otimes \dots \otimes S_r^{(k)}$, as desired.

The second part of the theorem comes from the fact that $\min_{i=1,\cdots,p} \prod_{j \neq i} d_j$ is a well known upper bound on the CP rank of a tensor of shape $d_1 \times d_2 \times \cdots \times d_k$ (see again (Kolda & Bader, 2009a)). $\square$

## A.2 Discussion on the Choice of Factorization

We consider various tensor factorization methods to decompose the high-order attention matrix:

- CP Decomposition: Based on the given proof, the CP decomposition of the attention matrix reshaped into a tensor of size $N_1^2 \times N_2^2 \times \cdots \times N_k^2$ is *equivalent* to the Kronecker factorization of the attention matrix used in HOT.

- Tucker Decomposition: The attention matrix could be expressed as a rank $R$ Tucker decomposition as:

$$
S_{\text{attention}} = \sum_{i_1,i_2,\dots,i_k}^{R} \mathcal{G}_{i_1,i_2,\dots,i_k} S_{i_1}^{(1)} \otimes S_{i_2}^{(2)} \otimes \cdots \otimes S_{i_k}^{(k)}, \tag{19}
$$

  where $S_{i_j}^{(j)}$ is the first-order attention matrix for axis $j$, and $\mathcal{G} \in \mathbb{R}^{R^K}$ is a core tensor that enhances expressivity. However, introducing the core tensor, would come at a cost of $\mathcal{O}(R^k)$ more parameters and a computational complexity of $\mathcal{O}(DR^k(\sum N_i) \prod N_i))$ growing exponentially with $R$, strongly limiting its scalability and efficiency.

- Tensor Train: For tensor train decomposition with a fixed rank for all factor matrices, the attention matrix is expressed as:

$$
S_{\text{attention}} = \sum_{i_1,i_2,\dots,i_{k-1}}^{R} S_{i_1}^{(1)} \otimes S_{i_1,i_2}^{(2)} \otimes S_{i_2,i_3}^{(3)} \otimes \cdots \otimes S_{i_{k-1}}^{(k)}, \tag{20}
$$

  where $S^{(i)} \in \mathbb{R}^{R \times N_i \times N_i}$ for $i \in \{1, k\}$ and $S^{(i)} \in \mathbb{R}^{R \times R \times N_i \times N_i}$ otherwise. Constructing fourth-order factor tensors $S^{(i)} \in \mathbb{R}^{R \times R \times N_i \times N_i}$ is non-trivial and may require additional parameters or modifications to align with our use case. This lack of simplicity makes tensor train decomposition less appealing as a design choice for our use case.

The choice of Kronecker decomposition in our work is motivated by its

---

[1]i.e., in pytorch $\mathcal{T}$ would be obtained obtained by permuting the modes of $\mathcal{S}$ and reshaping: torch.transpose($\mathcal{S}$, $[0, k+1, 1, k+2, \cdots, k-1, 2k-1]$.reshape($[N_1^2, \cdots, N_k^2]$)

1. Simplicity: Avoiding additional parameters, such as core tensors, making it easy to implement and interpret.

2. Efficiency: Scales efficiently with the tensor order, avoiding exponential growth in parameters or computations.

3. Expressivity and Performance: Comes with theoretical guarantee, while empirically performing well.

### A.3 Inductive Biases

The Higher-Order Transformer (HOT) introduces several key inductive biases that enhance its ability to process multiway data efficiently:

- **Preservation of Tensor Structure:** Unlike conventional Transformer models that require flattening or reshaping of input tensors into sequences, HOT inherently preserves the original multi-dimensional structure of the data. This is crucial for applications such as images, videos, and volumetric medical scans, where 2D or 3D neighborhood structures encode critical spatial and temporal relationships. By maintaining the tensor form, HOT enables direct modeling of multiway interactions without losing structural dependencies.

- **Locality and Spatial Awareness:** In many real-world datasets, relationships between neighboring elements in a tensor are often stronger than those between distant elements. This property, known as locality, is particularly evident in vision tasks, where adjacent pixels in an image or neighboring frames in a video exhibit high correlation. While self-attention mechanisms are inherently global, HOT retains weak locality by leveraging positional encodings, which helps the model differentiate between nearby and distant elements in the tensor space.

- **Global Interaction via Self-Attention:** Although locality is an important inductive bias, capturing long-range dependencies is equally crucial in high-dimensional data. Self-attention allows HOT to model global interactions across all tensor modes, enabling it to capture complex dependencies beyond local neighborhoods. This capability is particularly beneficial in tasks such as multivariate time-series forecasting, where dependencies between distant time steps play a key role in prediction accuracy.

- **Separable Structure through Kronecker Factorization:** A core inductive bias in HOT is the assumption that interactions across different tensor modes can be decomposed into separable components. This is achieved through Kronecker factorization, which models multiway dependencies in a structured and efficient manner. By assuming a factored representation of attention matrices, HOT significantly reduces computational complexity while still capturing meaningful relationships across dimensions.

## B Datasets Details

### B.1 Long-range Time-series Forecasting

We evaluate the performance of the proposed HOT model on five real-world datasets: Weather (Wu et al., 2021), consisting of 21 meteorological variables recorded every 10 minutes in 2020 at the Max Planck Biogeochemistry Institute, ECL (Wu et al., 2021), which tracks hourly electricity consumption for 321 clients, Traffic (Wu et al., 2021), collecting hourly road occupancy data from 862 sensors on San Francisco Bay area freeways between January 2015 and December 2016, and Solar-Energy (Lai et al., 2017), recording solar power production from 137 photovoltaic (PV) plants, sampled every 10 minutes in 2006.

We follow the data processing and train-validation-test split protocol used in TimesNet (Wu et al., 2023), ensuring datasets are chronologically split to prevent any data leakage. For forecasting tasks, we use a fixed lookback window of 96 time steps for the Weather, ECL, Solar-Energy, and Traffic datasets, with prediction lengths of 96, 192, 336, 720. Further dataset details are presented in Table 4.

Table 4: Timeseries forecasting dataset details.

| Dataset | Variables | Prediction Length | Train/Val/Test Size | Sample Frequency |
|---|---|---|---|---|
| Weather | 21 | {96, 192, 336, 720} | (36792, 5271, 10540) | 10min |
| ECL | 321 | {96, 192, 336, 720} | (18317, 2633, 5261) | Hourly |
| Traffic | 862 | {96, 192, 336, 720} | (12185, 1757, 3509) | Hourly |
| Solar-Energy | 137 | {96, 192, 336, 720} | (36601, 5161, 10417) | 10min |

## B.2 3D Medical Image Classification

We conduct experiments on the 3D subset of the Medical MNIST dataset (Yang et al., 2023). All datasets have an image size of $28 \times 28 \times 28$ voxels, allowing for consistent 3D image classification across different medical domains. The images come from various sources, ranging from human CT scans to animal microscopy, and have been adapted to create challenging classification tasks. Details are presented in Table 5.

- OrganMNIST3D is based on the same CT scan data used for the Organ{A,C,S}MNIST datasets, but instead of 2D projections, it directly uses the 3D bounding boxes of 11 different body organs. The dataset is adapted for a multiclass classification on organ identification from volumetric medical data.

- NoduleMNIST3D originates from the LIDC-IDRI dataset, a public repository of thoracic CT scans designed for lung nodule segmentation and malignancy classification. For this study, the dataset has been adapted for binary classification of lung nodules based on malignancy levels, excluding cases with indeterminate malignancy. The images are center-cropped and spatially normalized to retain a consistent voxel spacing.

- AdrenalMNIST3D features 3D shape masks of adrenal glands collected from patients at Zhongshan Hospital, Fudan University. Each shape is manually annotated by an expert endocrinologist using CT scans, though the original scans are not included in the dataset to protect patient privacy. Instead, the dataset focuses on binary classification of normal versus abnormal adrenal glands based on the processed 3D shapes derived from the scans.

- FractureMNIST3D is derived from the RibFrac dataset, which contains CT scans of rib fractures. The dataset classifies rib fractures into three categories (buckle, nondisplaced, and displaced), omitting segmental fractures due to the resolution of the images.

- VesselMNIST3D uses data from the IntrA dataset, which includes 3D models of intracranial aneurysms and healthy brain vessels reconstructed from magnetic resonance angiography (MRA) images. The dataset focuses on classifying healthy vessel segments versus aneurysms, with the models voxelized into 3D volumes.

Table 5: MedMNIST 3D datasets details.

| Dataset | Modality | Number of Classes | Train/Val/Test Size |
|---|---|---|---|
| Organ | Abdominal CT | 11 | (972, 161, 610) |
| Nodule | Chest CT | 2 | (1158, 165, 310) |
| Adrenal | Shape from Abdominal CT | 2 | (1188, 98, 298) |
| Fracture | Chest CT | 3 | (1027, 103, 240) |
| Vessel | Shape from Brain MRI | 2 | (1335, 192, 382) |

### B.3 Multispectral Image Segmentation

We evaluate our models on the SSL4EO-L Benchmark dataset (Stewart et al., 2023), a large-scale collection of multispectral Landsat images with corresponding land cover classification masks. For this work, we focus exclusively on images captured by the OLI/TIRS-RS sensor to ensure consistent spectral characteristics.

Each image has a resolution of $264 \times 264$ pixels (approximately 7.92 km $\times$ 7.92 km at 30 m/px) with 7 spectral bands, encompassing visible and infrared channels. The dataset includes 25,000 labeled images divided into training (20,000), validation (2,500), and test (2,500) splits, ensuring balanced data distribution.

The benchmark provides two versions of land cover annotations: the **NLCD version** with 17 broad land cover classes, and the **CDL version** with 134 finer-grained classes. Images are preprocessed to retain all multispectral channels, facilitating pixel-wise prediction of land cover classes in segmentation tasks. This setup allows us to evaluate HOT and baseline models in a controlled, consistent, and realistic multispectral segmentation setting.

### B.4 Implementation Details

Table 6: Hyperparameter Search Space.

| Hyperparameter | Value List |
|---|---|
| Number of Blocks (timeseries forecasting) | [2] |
| Number of Blocks (3D image classification) | [6] |
| Number of Blocks (multispectral image segmentation) | [6] |
| Number of Hidden Dimensions | [128] |
| Dropout | [0, 0.1, 0.2, 0.3] |
| Weight Decays | [0.01, 0.1, 0.3, 0.5] |
| Number of Attention Heads | [1, 2, 4, 8, 16, 32] |
| Patch Size | [2, 4] |
| Pooling Function | [Mean] |

**Timeseries Forecasting**  The convolution encoder is a single 1D convolution layer with kernel size and stride both set to patch size and applied on the temporal axis. This is equivalent to splitting the input timeseries into patches and applying a linear projection into the latent space of the model. Rotary positional encoding (Su et al., 2021) is applied only on the time axis. The output of the transformer is pooled before being fed to the final MLP layer by either taking the average over time. We conduct forecasting experiments by training models on each dataset. Following the same split of training/validation/test sets as in (Liu et al., 2024), the model weights from the epoch with the lowest MAE on the validation set are selected for comparison on the test set.

**3D Medical Image Classification**  The convolution encoder is implemented as a single-layer 3D convolution with a kernel size and stride equal to patch size. The output of the Transformer is pooled before being fed to the final MLP classifier by taking the average. We conduct classification experiments by training models on each dataset. We follow the official split of training/validation/test sets. The model weights from the epoch with the highest AUC score on the validation set are selected for comparison on the test set.

**Multispectral Image Segmentation**  The convolution encoder is implemented as a single-layer 2D convolution with a kernel size and stride equal to patch size. The output of the Transformer is directly fed to the final MLP classifier without pooling. We conduct classification experiments by training models on each dataset. We follow the official split of training/validation/test sets. The model weights from the epoch with the highest ACC score on the validation set are selected for comparison on the test set.

All the experiments are implemented in PyTorch and conducted on a single NVIDIA A100 GPU with 80 GB of memory, x86_64 CPU with 6 cores and 64GB of RAM. We utilize ADAM (Kingma & Ba, 2017) with an initial learning rate of $2 \times 10^{-4}$ and L2 loss for the timeseries forecasting task and cross-entropy loss for the medical image classification task. Our experiments show that using weight decay is crucial for avoiding overfitting in most cases. The batch size is uniformly set to 32, and the number of training epochs is fixed to 100. We conduct hyperparameter tuning based on the search space shown in Table 6.

## C   Baselines Details

### C.1   Timeseries Forecasting

We select seven Transformer-based models as benchmarks for our study, each offering unique approaches to time series forecasting:

- **Crossformer** (Zhang & Yan, 2023): This model explicitly captures dependencies across different dimensions in multivariate time series by embedding input data into a two-dimensional vector array and employing a Two-Stage Attention mechanism to model both temporal and cross-dimensional relationships efficiently.

- **Autoformer** (Wu et al., 2021): Autoformer introduces an Auto-Correlation mechanism to replace traditional self-attention, aiming to enhance the capture of long-term temporal dependencies while reducing computational complexity in time series forecasting.

- **FEDformer** (Zhou et al., 2022): FEDformer leverages a frequency-enhanced decomposition method to model time series data, focusing on extracting both trend and seasonal components to improve forecasting accuracy.

- **PatchTST** (Nie et al., 2023): PatchTST proposes a patching mechanism that partitions time series data into patches spanning multiple time steps, enabling the model to capture local temporal patterns and improve forecasting performance.

- **Transformer (spatial)** (Liu et al., 2024): This variation applies the attention mechanism along the spatial dimension, focusing on modeling dependencies between different variables at each time step.

- **Transformer (temporal)** (Liu et al., 2024): In contrast, this version applies attention along the temporal axis, emphasizing the modeling of temporal dependencies within each variable over time.

- **Transformer (non-factorized)**: This baseline employs the vanilla transformer architecture with a full self-attention mechanism over flattened input data, corresponding to non-factorized spatiotemporal attention. We consider this model as the main baseline to beat, as it models the full high-order attention.

### C.2   3D Medical Image Classification

We select seven medical image classifier models as baselines, each employing distinct methodologies for 3D medical image analysis:

- **ResNet-18/ResNet-50** (He et al., 2015; Yang et al., 2023): These convolutional neural networks (CNNs) utilize residual learning to facilitate the training of deeper architectures, effectively addressing the vanishing gradient problem. They have been widely adopted for various image classification tasks, including medical image analysis.

- **MDANet** (Huang et al., 2022): The Multi-Scale Discriminative Attention Network (MDANet) integrates multi-scale feature extraction with discriminative attention mechanisms to enhance the representation of critical regions in medical images, thereby improving classification performance.

- **CdTransformer** (Zhu et al., 2024): The Cross-Dimensional Transformer (CdTransformer) employs a cross-dimensional attention mechanism to capture both spatial and channel-wise dependencies in 3D medical images, facilitating more comprehensive feature learning for classification tasks.

- **ViT-3D** (Lai et al., 2024): This model extends the Vision Transformer (ViT) architecture to 3D medical image classification by dividing volumetric data into non-overlapping 3D patches and processing them using transformer encoders, effectively capturing global context without relying on convolutional operations.

- **ViViT-S** (Lai et al., 2024): The Video Vision Transformer (ViViT) adapts transformer architectures for video classification by modeling spatial and temporal dimensions jointly. The ViViT-S variant is a smaller-scale version designed to capture spatiotemporal features efficiently, making it suitable for 3D medical image classification tasks.

- **TimeSFormer** (Bertasius et al., 2021): TimeSFormer introduces a factorized attention mechanism that separately processes spatial and temporal dimensions, significantly reducing the computational complexity of standard full attention for video understanding. It achieves competitive performance on video benchmarks while being more memory-efficient than naive attention implementations.

- **Multiscale Vision Transformer** (Fan et al., 2021): MViT improves efficiency and scalability by progressively reducing spatial resolution while increasing channel capacity across layers. This multiscale design allows the model to capture hierarchical representations, making it suitable for tasks requiring fine-grained and coarse-grained feature integration.

- **Transformer (non-factorized)**: This baseline employs the vanilla transformer architecture with a full self-attention mechanism over flattened 3D medical image data, aiming to capture complex dependencies across all spatial dimensions without any factorization or dimensional reduction. We consider this model as the main baseline to beat as it utilizes the full high-order attention.

### C.3 Multispectral Image Segmentation

We select three models as baselines, each employing distinct methodologies:

- **ResNet-18/ResNet-50** (He et al., 2015)

- **ViT** (Dosovitskiy et al., 2020): ViT applies standard transformer architecture directly to image patches by flattening and embedding them as a sequence. While it achieves strong performance on image classification tasks, its full attention mechanism scales quadratically with the number of patches, leading to higher memory and computational requirements compared to factorized or hierarchical transformer variants.

## D  Additional Experiments

### D.1  Positional Encoding

In this section, we explore the influence of various positional encoding (PE) strategies on the performance of our proposed HOT model, focusing on the MedMNIST3D dataset. We evaluate 3D adaptations of widely-used PE methods, including sinusoidal encodings, learnable absolute positional encodings, and Rotary Positional Embeddings (RoPE). RoPE is applied mode-wise during the computation of the attention matrix for each corresponding mode, ensuring that positional information is effectively integrated into the model's attention mechanism.

The results reveal that the choice of positional encoding significantly impacts performance, with RoPE emerging as the optimal strategy for the Kronecker product variant of HOT. RoPE consistently achieves the highest accuracy and AUC scores across the Organ, Nodule, and Vessel datasets, with particularly strong results on the Organ dataset (99.8% AUC and 94.4% accuracy). In contrast, for the Kronecker sum variant,

Table 7: Ablations on different positional encoding strategies on 3D medical image classification task.

| Model | PE | Organ | | Nodule | | Vessel | |
|---|---|---|---|---|---|---|---|
| | | AUC | ACC | AUC | ACC | AUC | ACC |
| HOT (product) | NoPE | 99.2 | 88.5 | 87.2 | 85.5 | 80.7 | 89.3 |
| | 3D Absolute PE | 99.2 | 89.7 | 91.3 | 87.1 | 82.5 | **91.1** |
| | 3D Sin-Cos PE | 99.3 | 90.1 | 89.6 | 86.9 | **85.7** | 90.5 |
| | RoPE | **99.8** | **94.4** | **92.0** | **89.1** | **85.7** | 90.6 |
| HOT (sum) | NoPE | 99.2 | 89.2 | 87.9 | 86.1 | 79.8 | 90.1 |
| | 3D Absolute PE | 99.4 | 90.8 | **92.8** | **88.3** | 85.8 | **91.6** |
| | 3D Sin-Cos PE | 99.3 | 90.3 | 90.5 | 87.1 | **86.2** | 90.5 |
| | RoPE | **99.6** | **92.2** | 90.7 | 87.7 | 84.9 | 90.7 |

3D absolute learnable positional encoding proves to be the most effective in most cases, such as on the Nodule and Vessel datasets, where it achieves the highest AUC scores of 92.8% and 85.8%, respectively.

A key observation is that incorporating any form of positional encoding consistently improves performance compared to models without positional information NoPE. For instance, on the Vessel dataset, the use of 3D Sin-Cos PE boosts AUC by approximately 5% compared to NoPE. Notably, even without positional encodings, both HOT variants remain competitive with baseline models, demonstrating the robustness of the HOT framework. These findings underscore the importance of selecting appropriate positional encoding strategies tailored to the specific architecture and task, with RoPE and 3D absolute PE emerging as strong candidates depending on the model variant and dataset characteristics.

## D.2 Component Analysis

This section examines the contribution of key components—Attention and Feedforward Networks (FFN)—to the performance of the HOT model in time-series forecasting tasks. By systematically removing these components, we assess their individual and combined impact on forecasting accuracy across the Solar, Weather, and ECL datasets.

Table 8: Ablations on HOT components on timeseries forecasting task. We remove the components and assess the impact on the performance of HOT (product) model. The average results of all predicted lengths are listed here.

| Model | Components | | Solar | | Weather | | ECL | |
|---|---|---|---|---|---|---|---|---|
| | Attention | FFN | MSE | MAE | MSE | MAE | MSE | MAE |
| - | ✗ | ✓ | 0.240 | 0.269 | 0.265 | 0.283 | 0.193 | 0.288 |
| HOT (product) | ✓ | ✗ | 0.225 | 0.260 | 0.251 | 0.279 | 0.177 | 0.270 |
| | ✓ | ✓ | **0.221** | **0.257** | **0.245** | **0.275** | **0.169** | **0.268** |
| HOT (sum) | ✓ | ✗ | 0.230 | 0.272 | 0.250 | 0.282 | 0.190 | 0.288 |
| | ✓ | ✓ | 0.220 | 0.260 | 0.245 | 0.275 | 0.167 | 0.266 |

The results in Table 8 reveal that the attention module plays a pivotal role in the model's performance. Removing the attention module leads to a significant degradation in forecasting accuracy, with higher MSE and MAE values across all datasets. For instance, on the Solar dataset, the absence of attention increases

the MSE from 0.221 to 0.240, underscoring its importance in capturing temporal dependencies. In contrast, removing the FFN results in a comparatively smaller performance drop, suggesting that while the FFN contributes to the model's effectiveness, it is less critical than the attention mechanism. The best performance is achieved when both components are included, with the HOT (product) variant achieving the lowest MSE and MAE values across all datasets. For example, on the ECL dataset, the full model achieves an MSE of 0.169, compared to 0.193 when attention is removed.

These findings highlight the complementary roles of attention and FFN in the HOT framework. While attention is indispensable for modeling complex temporal relationships, the FFN provides additional refinement, contributing to the model's overall robustness and accuracy. This analysis underscores the importance of retaining both components to achieve optimal forecasting performance.

### D.3 Robustness Analysis

In this section, we evaluate the robustness of HOT by analyzing the stability of its performance across multiple runs with different random seeds. This analysis is conducted for both 3D medical image classification and time-series forecasting tasks, ensuring that the model's performance is consistent and reliable under varying initial conditions.

Table 9: Robustness of HOT performance on 3D medical image classification. The results are obtained from five random seeds.

| Variant | Organ | | Nodule | | Fracture | | Adrenal | | Vessel | |
|---|---|---|---|---|---|---|---|---|---|---|
| | AUC | ACC | AUC | ACC | AUC | ACC | AUC | ACC | AUC | ACC |
| Product | $99.8 \pm 0.4$ | $94.4 \pm 1.5$ | $92.0 \pm 0.5$ | $89.1 \pm 1.7$ | $73.6 \pm 0.3$ | $58.3 \pm 1.5$ | $88.6 \pm 0.2$ | $83.8 \pm 1.3$ | $85.7 \pm 0.7$ | $91.1 \pm 1.8$ |
| Sum | $99.6 \pm 0.3$ | $92.2 \pm 1.2$ | $90.7 \pm 0.6$ | $87.7 \pm 1.5$ | $74.9 \pm 0.3$ | $58.7 \pm 1.3$ | $87.3 \pm 0.5$ | $82.9 \pm 1.6$ | $84.9 \pm 0.6$ | $90.0 \pm 1.2$ |

Table 10: Robustness of HOT performance on timeseries forecasting. The results are obtained from five random seeds.

| Variant | ECL | | | Weather | | |
|---|---|---|---|---|---|---|
| | MSE | MAE | SMAPE | MSE | MAE | SMAPE |
| Product | $0.169 \pm 0.002$ | $0.268 \pm 0.003$ | $0.520 \pm 0.003$ | $0.245 \pm 0.004$ | $0.275 \pm 0.003$ | $0.642 \pm 0.005$ |
| Sum | $0.167 \pm 0.003$ | $0.266 \pm 0.002$ | $0.516 \pm 0.003$ | $0.245 \pm 0.005$ | $0.275 \pm 0.004$ | $0.637 \pm 0.003$ |

| Variant | Traffic | | | Solar | | |
|---|---|---|---|---|---|---|
| | MSE | MAE | SMAPE | MSE | MAE | SMAPE |
| Product | $0.420 \pm 0.004$ | $0.278 \pm 0.006$ | $0.515 \pm 0.005$ | $0.221 \pm 0.002$ | $0.257 \pm 0.003$ | $0.420 \pm 0.004$ |
| Sum | $0.428 \pm 0.003$ | $0.280 \pm 0.005$ | $0.530 \pm 0.004$ | $0.220 \pm 0.002$ | $0.260 \pm 0.002$ | $0.416 \pm 0.003$ |

For 3D medical image classification (Table 9, both HOT (product) and HOT (sum) exhibit stable performance across five runs, with low standard deviations in AUC and accuracy metrics. For example, on the Organ dataset, HOT (product) achieves an AUC of $99.8 \pm 0.4$, while HOT (sum) maintains an AUC of $99.6 \pm 0.3$. In time-series forecasting (Table 10, both variants consistently achieve low error rates with minimal variability, as evidenced by SMAPE, MSE, and MAE metrics. On the ECL dataset, HOT (product) achieves an MSE of $0.169 \pm 0.002$, and HOT (sum) achieves an MSE of $0.167 \pm 0.003$.

The robustness analysis confirms that HOT delivers stable and reliable performance across both medical imaging and time-series forecasting tasks. Its low variability in key metrics, combined with strong predictive accuracy, makes it a dependable choice for real-world applications where consistency is critical.

## D.4 Visualization of Attention Maps

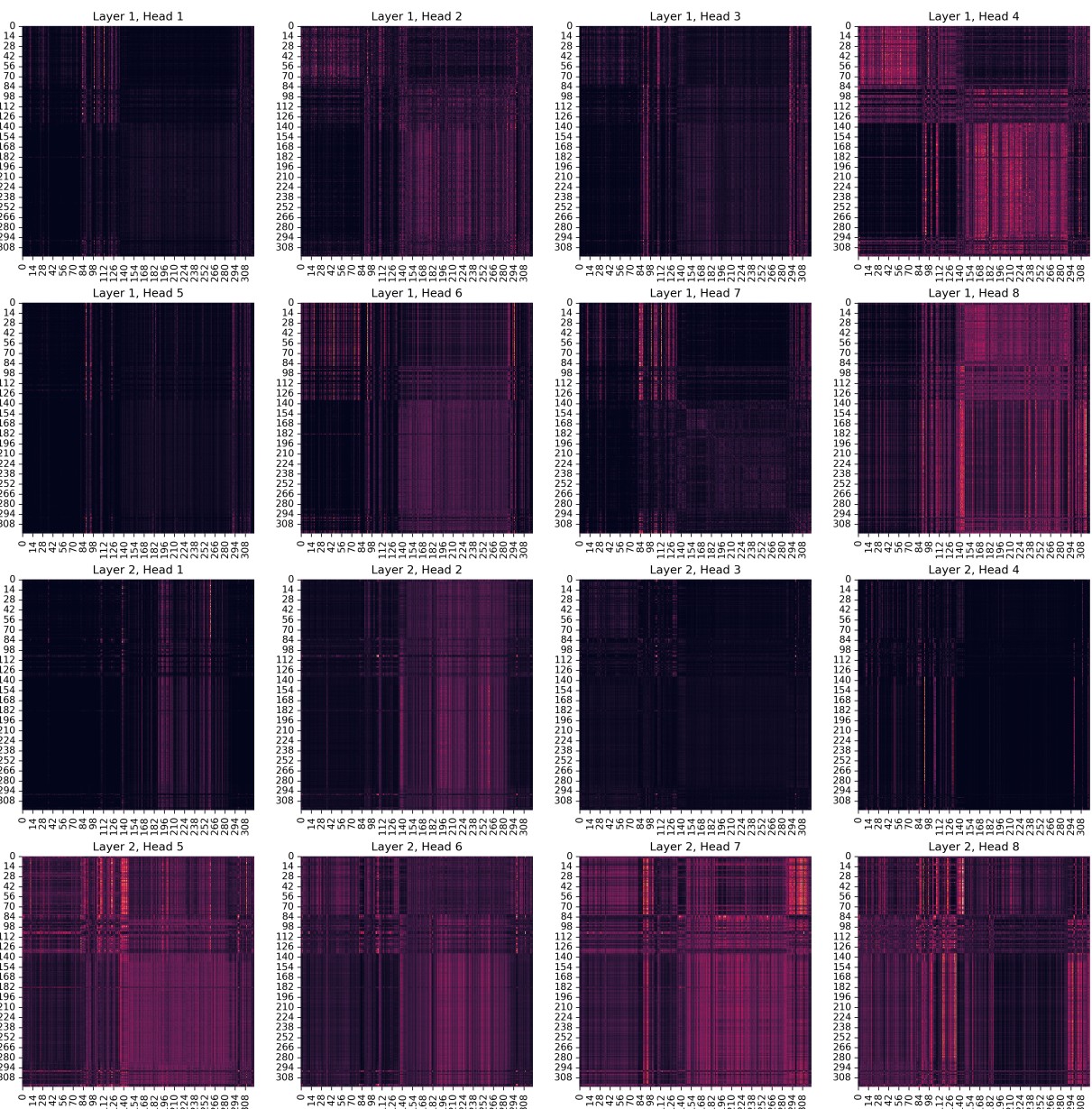

Figure 10: Visualization of the spatial attention maps learned by HOT (product) on ECL dataset.

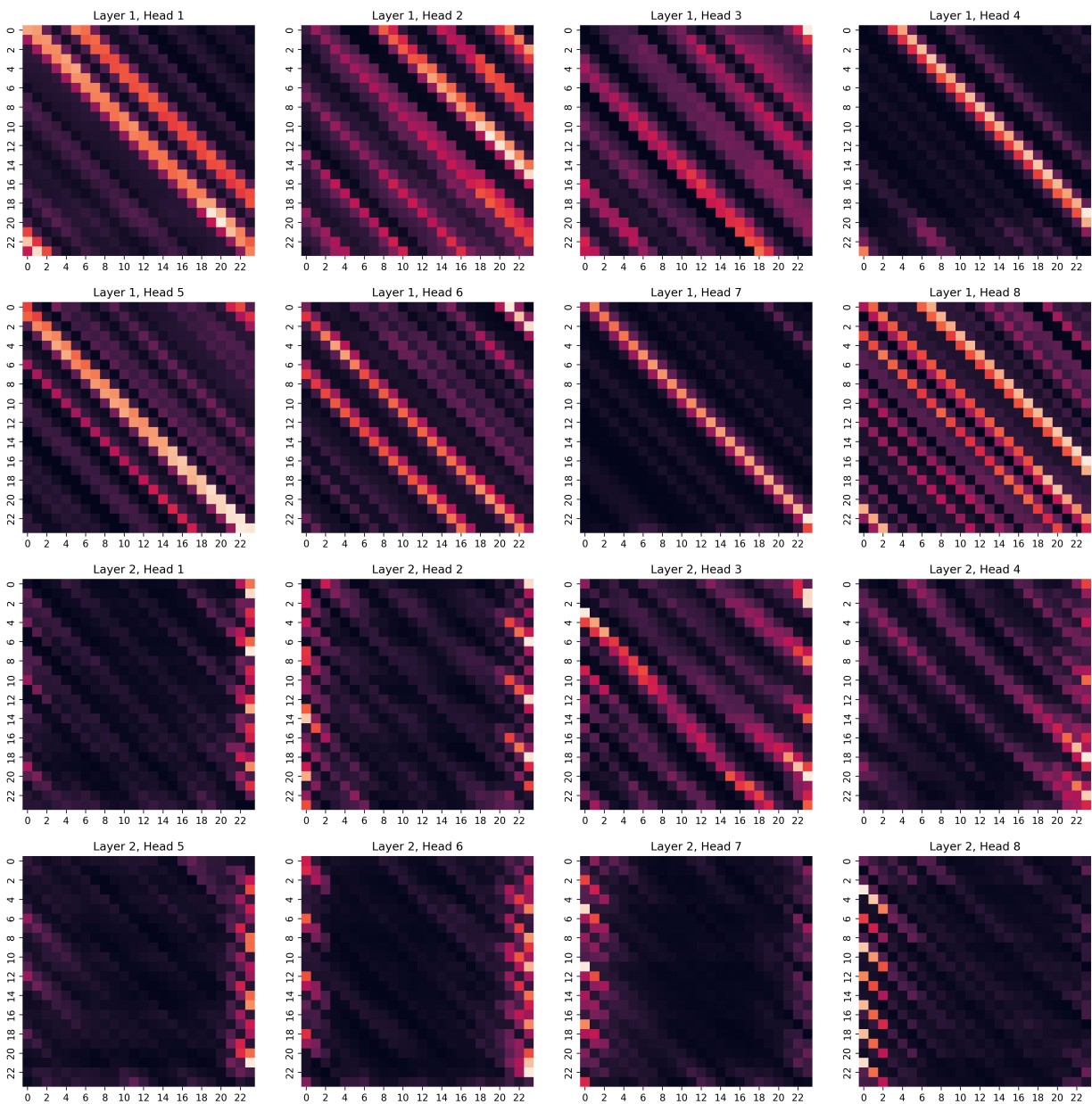

Figure 11: Visualization of the temporal attention maps learned by HOT (product) on ECL dataset.

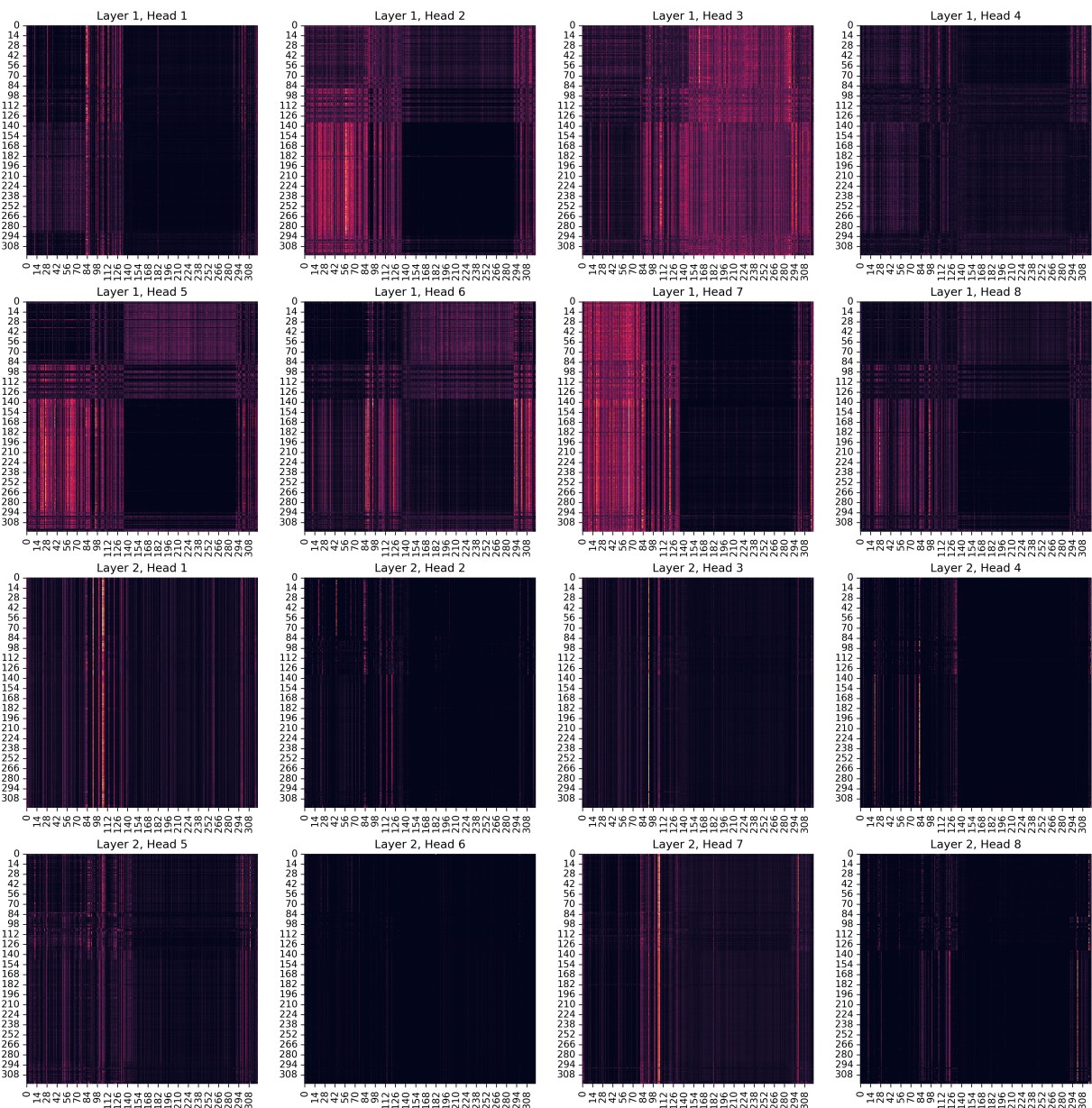

Figure 12: Visualization of the spatial attention maps learned by HOT (sum) on the ECL dataset.

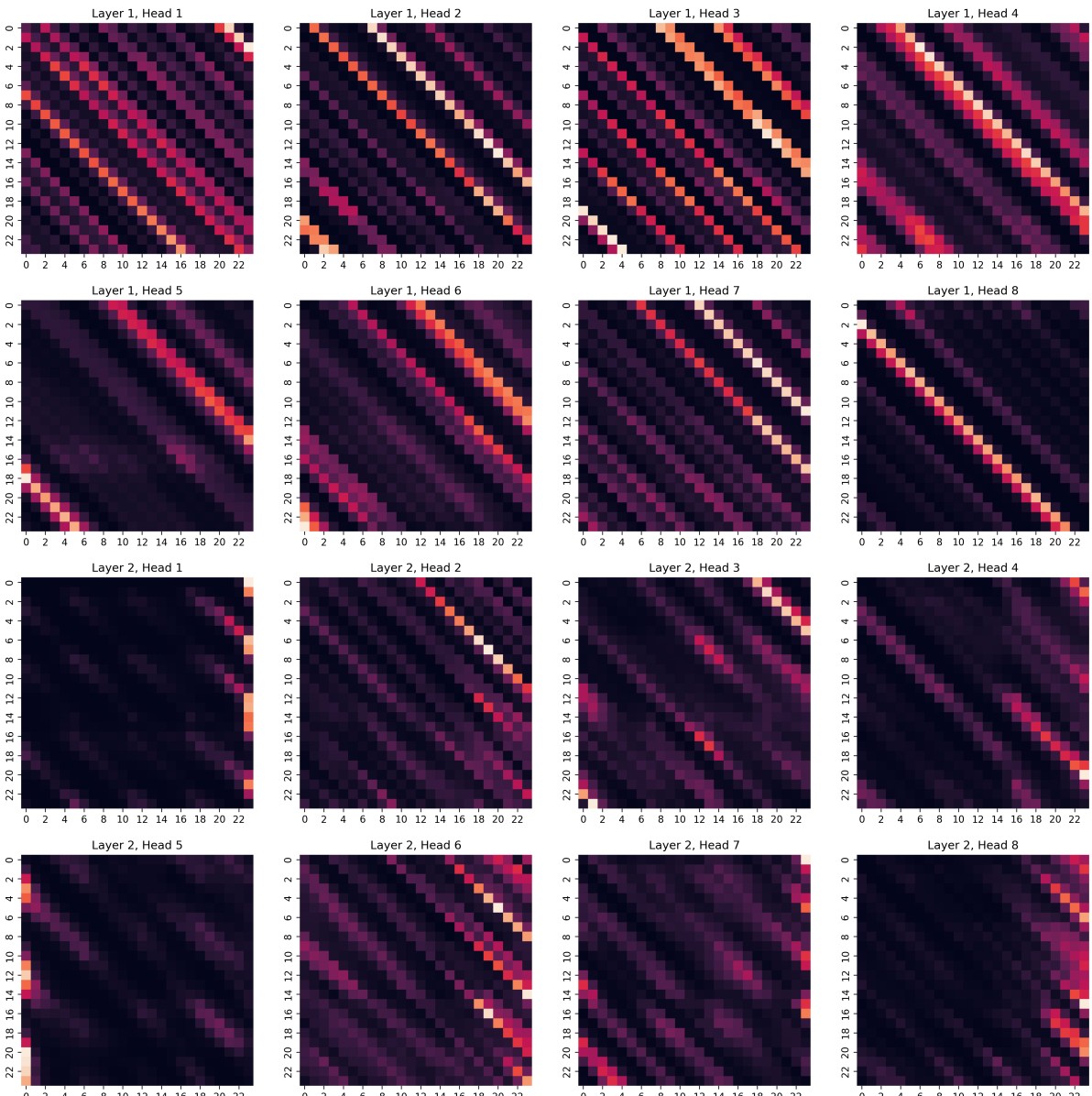

Figure 13: Visualization of the temporal attention maps learned by HOT (sum) on the ECL dataset.

