# OpenReview forum: "Higher Order Transformers With Kronecker-Structured Attention"
_TMLR — Accepted by TMLR_

### Review · Reviewer_nCnc · 2025-08-20

**Summary Of Contributions:**

This paper aims to propose an alternative attention mechnisim which replaces the standard self-attention on high-dimensional data, e.g. 3D data, with Kronecker-Structured attention. With that, the proposed method can reduce the training time and memory usage significantly while retaining the competitive performance. The proposed method is evaluated via two types of datasets, time-series prediction and classification tasks.

Strengths:

1. The proposed attention method reduces the training time and memory usage significantly, which opens the potential to scale the model size with reasonable training time and hardware resources.

2. The derivation of HOT is clear and easy to follow, and the support of theoretical analysis on the design of HOT is good.

Weaknesses:
1. For the experiments on 3D data classification, there is no comparison to other efficient attention methods, that are applied on the similar data, e.g. video.

2. Even though the paper included empirical savings in training time and memory usage, there is no analytic analysis how the savings on FLOPs with respect to data size;  moreover, there is no comparison on inference time.

**Audience:**

Yes

**Audience Explanation:**

The readers will be interested to learn more about efficient attention and learning how to apply it to 3D data.

**Claims And Evidence:**

Yes

**Claims Explanation:**

The paper provided theoretical analysis on the proposed HOT and evaluated on two types of datasets.

**Requested Changes:**

1. See weakeness.

2. The paper lacks the comparison to those 3D attention methods that are used in video modeling as the video is also a part of 3D data, e.g. TimeSformer [1] and MViT [2].
Reference:
[1] Is Space-Time Attention All You Need for Video Understanding?
[2] Multiscale Vision Transformers

3. Any insight that the proposed method could work on video recongition?

4. For section 5.3.2, doesn't the number of attention heads also affect the vanilla attention? Is that phonemonon uniqle to the proposed HOT? It will be great to include the results of original attnetion.

---

> ### Author Response · Authors · 2025-09-18
>
> We thank the reviewer for their detailed and constructive feedback. We are pleased that you found both the effectiveness and efficiency of our method valuable. Below, we address your concerns and outline the improvements incorporated in the revised version:
>
> - **Additional baseline comparisons:** We report results for MViT and TimeSFormer on the MedMNIST3D dataset. Both models perform worse than HOT across all 3D benchmarks, highlighting HOT’s advantages in 3D tasks compared to other efficient attention methods.
>
> - **Runtime comparisons:** We updated Figure 7 with a comprehensive efficiency analysis of HOT vs ViT and ResNet over training/inference time and memory footprint. We also added a new plot in Figure 6 illustrating the effect of input data size on FLOPS, to further demonstrate HOT’s computational efficiency compared to vanilla transformer and TimeSformer.
>
> - **Video Recognition:** As mentioned in the conclusion, exploring HOT’s capabilities and computational gains on other modalities, such as video, remains a promising direction for future work.
>
> - **Number of Heads:** Following common practice, we treat the number of attention heads as a hyperparameter and select the best value via grid search for both HOT and vanilla transformer (non-factorized). Figure 5 has been updated to include results for the vanilla non-factorized transformer as well.
>
> We hope these clarifications address your comments thoroughly and greatly appreciate your thoughtful engagement with our work.

---

### Review · Reviewer_rjzJ · 2025-08-21

**Summary Of Contributions:**

This paper introduces HOT, a higher order Transformer architecture using Kronecker products (sums).
The approach is theoretically introduced and tested on a couple of datasets showing good results as well as efficiency improvements.

**Audience:**

Yes

**Audience Explanation:**

Higher order Attention for Higher order inputs is of high practical and theoretical relevance

**Claims And Evidence:**

Yes

**Claims Explanation:**

The paper is - for the methodology and notation section - nicely written. I am not particularly an expert in this area, but the proofs and concepts check out and are consistent

**Requested Changes:**

# Strength
- Overall, I think its a nice approach with really concise and nice presentation of the methodology
- It is, to the best of my knowledge a novel idea
- Very good preliminaries section with concise explanations for unfamiliar readers

# Weaknesses
- Throughout the paper there are several typos and grammatical issues. I'd ask the authors to do a proper and careful reading to smooth out all of the rough edges
- Do I understand it correclty, that you perform hpo for the presented model, but not for the baselines and that experimental results are obtained by a single run?
    - Why are the number of blocks and hidden dimensions fixed over HPO runs? Is it a randomized search then over different dropout, weight decay, attention heads and patch size parameters?
- For runtime analysis please provide all details, i.e. system, GPU, epochs, model size etc.


# Minor/Questions
- Not sure, whether (KAN) is the best abbreviation. I know KAN as Kolmogorov Arnold Networks, but
- Do I understand it correctly in section 4.1 that this is just flattening + reshaping? Also, maybe it could be made clearer that this is not your suggested approach, but something that is commonly used? It's rather unprominently written in the last sentence of the paragraph
- The bullet points contributions in the beginning are rather a poor representation of the actual contributions.
- Reporting MSE and MAE seems a bit redundant for the forecasting table
-

---

> ### Author Response · Authors · 2025-09-18
>
> We thank the reviewer for their positive assessment of our work and their thoughtful comments. We are pleased that you found both the effectiveness and efficiency of our method valuable. Below, we address the points you raised and provide clarifications to further strengthen the paper:
>
> - **Text errors and typos:** We appreciate the feedback and we have corrected all text errors and typos in the revision.
>
> - **Baseline results and hyperparameter optimization:** Some baseline results are reported from their original papers, where the authors performed hyperparameter optimization (HPO) to obtain the best results. We have updated the tables to clearly indicate which results are taken from the original papers. For the rest of the results, we followed the same procedure and performed a grid search over the full hyperparameter space outlined in Table 6.
>
> - **System details:** Hardware system details are included in Section 5.4.3 and the appendix B.4 in the revision for improved presentation and accessibility.
>
> - **Naive high-order attention:** As noted, Section 4.1 describes the common and naive implementation of high-order attention, which is equivalent to flattening and reshaping the tensor. We have clarified this point in the revision of this section.
>
> - **Contributions list:** We have updated the introduction to include a finer-grained and more detailed list of contributions in the revised version.
>
> - **Reporting metrics:** We understand that reporting both MSE and MAE may seem redundant. However, we follow standard practice in the time-series forecasting literature to maintain consistency with prior references and baseline comparisons.
>
> We hope these clarifications address your comments thoroughly, and we greatly appreciate your thoughtful engagement with our work.

---

### Review · Reviewer_WHxp · 2025-09-03

**Summary Of Contributions:**

This paper proposes Higher-Order Transformers (HOT), a generalized Transformer architecture designed to efficiently handle high-dimensional tensor data by factorizing the attention mechanism using Kronecker products and sums. The key contribution is decomposing the computationally expensive full attention matrix (which scales quadratically with tensor size) into mode-wise attention matrices that can be combined through Kronecker operations, reducing complexity from O(D(N₁N₂...Nₖ)²) to O(D(∑ᵢNᵢ)(∏ⱼNⱼ)) while preserving tensor structure and cross-dimensional dependencies, where D is the embedding dimension, and N₁N₂...Nₖ are the sizes of tensors. The authors provide theoretical guarantees showing that their factorized approach can approximate any high-order attention matrix with sufficient rank, and demonstrate competitive performance on multivariate time series forecasting and 3D medical image classification tasks with significantly reduced computational and memory costs compared to full attention mechanisms.

**Audience:**

Yes

**Audience Explanation:**

This paper addresses a fundamental computational bottleneck that affects multiple active research areas in machine learning. The quadratic scaling of attention mechanisms has been a critical challenge limiting the application of Transformers to high-dimensional data, and this work provides a principled solution through tensor factorization that maintains theoretical guarantees while achieving substantial practical speedups. The approach is particularly relevant to researchers working on video analysis, 3D medical imaging, climate modeling, and multivariate time series - domains where data naturally exists in tensor form but current methods either flatten the structure (losing important relationships) or face prohibitive computational costs. I believe that the broader TMLR community would likely find value in both the theoretical contributions and practical implications.

**Broader Impact Concerns:**

The paper demonstrates HOT on 3D medical imaging classification but doesn't address the critical ethical implications of deploying AI models in healthcare.

**Claims And Evidence:**

Yes

**Claims Explanation:**

Strengths:
1. The paper provides rigorous theoretical analysis including universality guarantees (Theorem 4.3) showing that Kronecker decomposition can represent any high-order attention matrix, along with stability analysis of the row-stochastic properties and rank bounds.
2. HOT achieves substantial reductions in memory usage (40-60%) and training time while maintaining competitive performance, making it practical for resource-constrained applications with high-dimensional data.

Weaknesses:
1. The evaluation is restricted to relatively small-scale datasets (28×28×28 medical images, time series with modest dimensions), leaving questions about scalability to truly large tensor data that would most benefit from this approach.
2. While the paper proves universal approximation capability, it lacks detailed analysis of how well low-rank Kronecker decompositions approximate full attention in practice, and doesn't provide clear guidance on selecting the factorization rank for different data characteristics.

**Requested Changes:**

1. The experimental evaluation is limited to relatively small tensors (28×28×28 medical images, modest time series dimensions). The paper would benefit from experiments on larger-scale data where the computational advantages would be most pronounced - such as high-resolution video analysis, large spatiotemporal climate datasets, or higher-dimensional medical volumes.
2. While Theorem 4.3 proves universal approximation capability, the paper lacks analysis of how approximation quality degrades with different factorization ranks in practice. Adding empirical studies showing the trade-off between rank and approximation error would provide clearer guidance for practitioners.
3. Some complexity expressions omit the hidden dimension D. The paper should consistently include D in all complexity analyses for accuracy.

---

> ### Author Response · Authors · 2025-09-18
>
> We thank the reviewer for their detailed and constructive feedback. We are pleased that you found both the effectiveness and efficiency of our method valuable. Below, we address your concerns and outline the improvements incorporated into the revised version:
>
> 1. **Scalability to larger tensors:** The Traffic dataset already consists of relatively large tensor inputs ($862 \times 96$). In the revision, we additionally report results on the SSL4EOL Benchmark dataset for pixel-wise classification on multispectral satellite images, where inputs have size $264 \times 264 \times 7$. These results further demonstrate the merits of our method: HOT outperforms ViT and performs on par with ResNet variants, while requiring only a fraction of the parameters and computation. We also added an efficiency analysis in Figure 6 on the effect of input data size on FLOPS, to further demonstrate HOT’s computational efficiency on large scale tensors.
>
> 2. **Factorization rank and approximation quality:** As correctly noted, the approximation quality of the attention module depends on the factorization rank. The impact of factorization rank on final performance is analyzed in Section 5.3.2. Following common practice, we treat the rank (i.e., number of attention heads) as a hyperparameter and determine the best value through hyperparameter search. This clarification has been added to the revision.
>
> 3. **Complexity expression:** We thank the reviewer for pointing out the missing hidden dimension $D$ in the complexity expression in Section 4.1. This has been corrected in the revised version.
>
> 4. **Broader impact statement:** We included the following statement at the end of the revised paper:
>  > This work introduces Higher-Order Transformers (HOT) as an efficient and interpretable framework for modeling multiway data, with experiments on 3D medical imaging benchmarks. Potential benefits include faster and more accurate image analysis, lower computational costs that broaden access, and improved transparency through mode-wise attention maps. However, risks include bias from non-diverse datasets, limited generalization, privacy concerns, and automation bias in clinical use. Since our evaluations are limited to research benchmarks, real-world performance and interpretability remain uncertain. Future work should focus on fairness audits, robust validation on diverse clinical data, uncertainty quantification, and human-in-the-loop deployment to ensure safe and ethical use.
>
> We hope these clarifications address your comments thoroughly, and we greatly appreciate your thoughtful engagement with our work.

---

### Decision · Action_Editor_5bBb · 2025-10-22

**Recommendation:** Accept as is

**Audience:**

Yes

**Audience Explanation:**

Improving the efficiency of attention mechanisms is an important research topic, as attention-based architectures (e.g., Transformers) have been widely adopted across various domains. As all reviewers noted, the TMLR community would likely be interested in the findings of this submission.

**Claims And Evidence:**

Yes

**Claims Explanation:**

As all the reviewers noted, the claims made in this paper are well supported by several theoretical justifications and experimental results.